# Oxidized hemoglobin triggers polyreactivity and autoreactivity of human IgG via transfer of heme

Cyril Planchais[1,3], Remi Noe [2,3], Marie Gilbert[2], Maxime Lecerf [2], Srini V. Kaveri[2], Sébastien Lacroix-Desmazes[2], Lubka T. Roumenina [2] & Jordan D. Dimitrov [2✉]

Intravascular hemolysis occurs in diverse pathological conditions. Extracellular hemoglobin and heme have strong pro-oxidative and pro-inflammatory potentials that can contribute to the pathology of hemolytic diseases. However, many of the effects of extracellular hemoglobin and heme in hemolytic diseases are still not well understood. Here we demonstrate that oxidized hemoglobin (methemoglobin) can modify the antigen-binding characteristics of human immunoglobulins. Thus, incubation of polyclonal or some monoclonal human IgG in the presence of methemoglobin results in an appearance of binding reactivities towards distinct unrelated self-proteins, including the protein constituent of hemoglobin i.e., globin. We demonstrate that a transfer of heme from methemoglobin to IgG is indispensable for this acquisition of antibody polyreactivity. Our data also show that only oxidized form of hemoglobin have the capacity to induce polyreactivity of antibodies. Site-directed muta-genesis of a heme-sensitive human monoclonal IgG1 reveals details about the mechanism of methemoglobin-induced antigen-binding polyreactivity. Further here we assess the kinetics and thermodynamics of interaction of a heme-induced polyreactive human antibody with hemoglobin and myoglobin. Taken together presented data contribute to a better under-standing of the functions of extracellular hemoglobin in the context of hemolytic diseases.

[1] Laboratory of Humoral Immunology, Institut Pasteur, Université Paris Cité, INSERM U1222, 75015 Paris, France. [2] Centre de Recherche des Cordeliers, INSERM, CNRS, Sorbonne Université, Université Paris Cité, 75006 Paris, France. [3] These authors contributed equally: Cyril Planchais, Remi Noe. ✉email: jordan.dimitrov@sorbonne-universite.fr

Many pathological conditions are associated with an uncontrolled destruction of the red blood cells. As the principal constituent of erythrocytes is hemoglobin (Hb), hemolysis leads to a liberation of massive quantities of this hemoprotein in the plasma. The extracellular Hb is spontaneously oxidized to metHb, a process associated with a generation of reactive oxygen species[1,2]. The oxidation of Hb facilitates dissociation of its prosthetic group heme (iron protoporphyrin IX) from the globin chains. Free Hb and heme have prominent pathogenic potentials. Hb can scavenge nitric oxide and thus interfere with regulation of the vascular tone[3–6]. Hb and heme are also pro-oxidative, cytotoxic, and pro-inflammatory[6–14]. It has been demonstrated that heme can activate distinct types of innate immune cells and endothelium, thus triggering pathological inflammatory reactions[15–25]. The extracellular heme has also been demonstrated to affects the functions of different plasma proteins[26]. Thus, heme can activate the alternative pathway of the complement system, and interferes with blood coagulation and thrombosis[27–39]. Others and we have shown that heme has a prominent potential to modify the antigen-binding characteristics of some immunoglobulins. Thus, upon exposure to heme, a fraction of antibodies in human immune repertoires, irrespectively of their immunoglobulin class, acquire polyreactivity and start to recognize a large panel of self- or pathogen-derived antigens with substantial binding affinities ($K_D$ value in low nanomolar range)[40–47]. Importantly, a recent study showed that some clinical-stage therapeutic monoclonal antibodies can also acquire polyreactivity after contact with heme[48].

The ability of heme to modulate the antigen-binding characteristics of antibodies is dependent on direct binding of the cofactor molecule to the variable region. It was found that the heme sensitive antibodies are characterized with particular sequence traits of their variable regions, i.e., they possess significantly higher number of aromatic and positively charged amino acid residues in their CDR loops[47,48]. Nevertheless, the molecular mechanism responsible for the heme-induced polyreactivity of antibodies is ill defined. The biological role of this phenomenon is also not well understood. Importantly, immunoglobulins that were in vitro exposed to heme demonstrated protective effects in animal models of systemic inflammation and autoimmunity[49,50]. These results suggest that the heme-sensitive antibodies may be endowed with an immune-regulatory potential. However, as heme induces appearance of reactivity to self-antigens such antibodies may also have some pathogenic activity especially in the context of hemolytic diseases. Moreover, heme-induced changes in antigen-binding reactivity of therapeutic antibodies can negatively impact their therapeutic activity.

Heme is a downstream product of the hemolysis. Under hemolytic conditions immunoglobulins (and other plasma proteins) will first experience the effects of extracellular Hb. Hb is characterized by sophisticated redox chemistry[1,9,10,14]. Hb can cause oxidation of proteins, lipids, and nucleic acids[2,51,52]. Therefore, this protein should be rapidly removed from the circulation[6,53,54]. The principle Hb scavenger in the plasma is haptoglobin[6,55,56]. This protein tightly binds Hb and delivers it (through interaction with CD163) to macrophages for degradation[57]. It has been reported, however, that in hemolytic conditions the plasma pool of haptoglobin is severely depleted[58]. In such cases a high level of Hb (20–50 μM) can be maintained in the blood plasma for extended periods of time (>24 h). This suggests that plasma macromolecules might experience effects of Hb and its prosthetic group heme. Although there are experimental data about the effect of heme on immunoglobulins, to the best of our knowledge it is still not investigated how an early component of hemolysis, such as Hb, could impact functions of human antibodies. These data would be a valuable piece of information for understanding of the physio-pathological consequence of induction of polyreactivity of antibodies under hemolytic conditions.

In the present study we investigated the functional impact of Hb on human immunoglobulins. We demonstrate that incubation of polyclonal or some monoclonal human IgG antibodies in the presence of oxidized hemoglobin (methemoglobin) resulted in an appearance of polyreactivity and reactivity towards distinct self-proteins. This observation may partly explain elevated antibody autoreactivity that accompanies some hemolytic disorders.

## Results

**Oxidized Hb induces polyreactivity and autoreactivity of human IgG.** Previous studies have demonstrated that the exposure of human pooled IgG to heme results in a gain of antibody reactivity to multiple unrelated protein antigens and to phospholipids[40,43]. Here we tested whether Hb has an impact on antigen binding characteristics of human antibodies. As target antigens, we used a set of structurally unrelated human proteins—factor VIII (FVIII), factor IX (FIX) and apo-Hb. Presumably antibody repertoires of healthy individuals should not display reactivities towards these proteins. Incubation of a normal pooled human IgG with oxidized hemoglobin, i.e., metHb ($Fe^{3+}$ heme), resulted in a concentration-dependent binding of IgG to the studied proteins (Fig. 1a). The reactivity of native pooled IgG to these proteins was only marginal. Oxidized Hb was also able to induce the binding of human IgG towards many proteins that are present in a human endothelial cell lysate, as observed by immunoblot analyses (Fig. 1b). Size exclusion chromatography analyses of IgG exposed to metHb indicated that the observed induction of autoreactivity of antibodies following exposure to metHb was not a result of changes in overall molecular composition of IgG preparation (Supplementary Fig. 1). The capacity of metHb to induce binding of human antibodies to its protein component (apo-Hb or globin) could explain the fact that at the highest concentrations of metHb used for treatment of IgG, there was a slight decrease in the reactivity towards surface-immobilized proteins (Fig. 1a). The tendency of metHb to induce IgG reactivity to globin was also confirmed by fluorescence spectroscopy analyses. Indeed, the binding of a fluorescence probe—8-anilinonaphthalene-1-sulfonate (ANS)—that specifically associates (and as result increases its fluorescence) with hydrophobic patches on proteins, revealed that total free hydrophobic sites on proteins were reduced by mixing human IgG and human metHb. Thus, the observed increase in the fluorescence signal of ANS of mixture of IgG and metHb was lower as compared to the increase that results from the sum of the fluorescence detected when the proteins were tested individually (Supplementary Fig. 2). These data suggested that IgG interaction with metHb results in burial of some hydrophobic patches on the surface of the proteins.

Further, we investigated the binding behavior of a human monoclonal IgG1, Ab21 that was previously shown to bind heme and acquire polyreactivity following this interaction[59]. Incubation of this monoclonal antibody with metHb lead to a substantial gain of binding to endothelial cell antigens (Fig. 1c) and to a number of unrelated proteins and LPS (Fig. 1d). In other words, the monoclonal antibody acquired antigen-binding polyreactivity. The effect of metHb on the antigen-binding reactivity of the Ab21 was concentration-dependent as observed by immunoblot analyses (Fig. 1c). Interestingly, at low concentrations metHb (IgG:Hb molar ratios of 100:1) was also able to uncover considerable polyreactivity of this monoclonal IgG1.

By using ELISA, we next investigated effect of other hemoproteins on the reactivity of human IgG. As in the case of

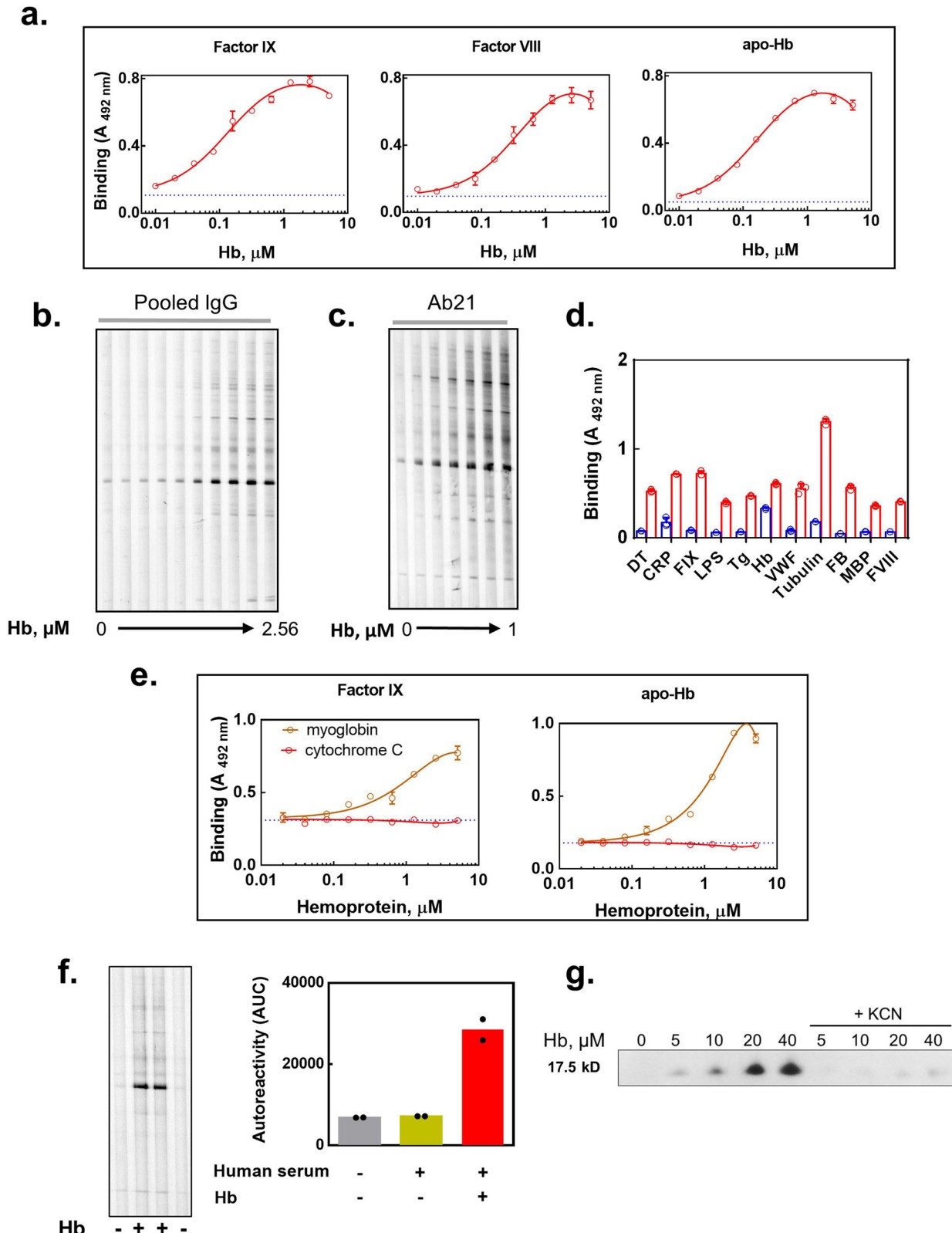

metHb, incubation of the pooled human IgG with myoglobin resulted in a concentration-dependent increase in the binding to immobilized human proteins. In contrast, cytochrome C in the same concentration range did not modify the specificity of antibodies (Fig. 1e).

To assess whether metHb affects IgG binding characteristics in a more physiological context, we incubated a healthy human serum with metHb. As depicted in Fig. 1f, this incubation resulted in an augmentation of the total autoreactivity of IgG towards endothelial cell antigens. Moreover, addition of metHb to a normal human serum caused formation of IgG/Hb complexes, as revealed by co-precipitation experiments using protein G beads (Fig. 1g). Cyanide anions represent a high affinity ligand for heme's iron ion. Their interaction with heme results to

**Fig. 1 Oxidized Hb induces polyreactivity and autoreactivity of human antibodies. a** ELISA assessment of binding of human pooled IgG to immobilized factor IX, factor VIII, and apo-hemoglobin. Pooled IgG at 10 μM was first pre-incubated for 30 min with increasing concentrations of Hb (0, 0.01–5.12 μM) and then incubated with immobilized proteins for 2 h at concentration of 1 μM. Each point represents mean absorbance ±SD from $n = 4$ repetitions of each sample. The dashed line represents the reactivity of native IgG preparation. A representative result from three independent experiments is shown. **b** Immunoblot analyses of the reactivity of human pooled IgG towards proteins from human endothelial cells. The pooled IgG (10 μM) was pre-incubated with increasing concentrations of Hb (0; 0.01–2.56 μM) for 30 min. For assessment of the immunoreactivity IgG were diluted to 1 μM and incubated with membrane-bound endothelial proteins for 2 h. **c** Immunoblot analyses of the reactivity of monoclonal human IgG1 Ab21 towards proteins from human endothelial cells. Ab21 (1 μM) was pre-incubated with increasing concentrations of metHb (0; 0.025, 0.05, 0.1, 0.25, 0.5, and 1 μM) for 30 min. For assessment of the immunoreactivity, the antibody was diluted to 0.1 μM and incubated with membrane-bound endothelial proteins for 2 h. **d** Binding of Ab21 to a panel of protein antigens and LPS. The antibody was first pre-incubated at 5 μM with 1 μM of metHb, then it was diluted to 0.5 μM and incubated with immobilized antigens for 2 h. The blue bars depict the antigen binding by native Ab21, the red bars depict the binding by Hb-exposed Ab21. Each bar represents mean absorbance ±SD from $n = 3$ repetitions of each sample. The individual replicates were depicted as circles. **e** Induction of autoreactivity of human antibodies by hemoproteins. Binding of human pooled IgG to immobilized factor IX and apo-Hb. Pooled IgG at 10 μM was first pre-incubated for 30 min with increasing concentrations of myoglobin or cytochrome c (0, 0.01–5.12 μM). After samples were diluted 10× and incubated for 2 h with immobilized proteins. Orange line depicts immunoreactivity of IgG after incubation with myoglobin; red line, depicts immunoreactivity after incubation with cytochrome c. Each point represents mean absorbance ±SD from $n = 3$ repetitions of each sample. The dashed lines represent the reactivity of native IgG. A representative result from two independent experiments is shown. **f** Immunoblot analyses of the reactivity of human IgG in intact serum towards proteins from endothelial cells. To normal human serum diluted 1:1 with PBS was added metHb to final concentration of 50 μM. Following incubation for 2 h, the serum was diluted 100× and incubated with immobilized proteins for 2 h. The bar graph depicts the mean area under the curves obtained after quantification of immunoreactivity in two lines for each condition. The individual replicates were depicted as black circles. **g** Immunoprecipitation of metHb from healthy human serum. metHb or metHb pre-incubated with KCN (5 mM) were added to normal human serum resulting in final concentrations of 5, 10, 20, and 40 μM. After incubation for 2.5 h at 37 °C, serum IgG was purified by protein G magnetic beads. Presence of immune complexes of IgG with metHb was detected by anti-human Hb goat polyclonal antibody. Immunoreactivity was evaluated by chemiluminescence analyses.

displacement of other heme ligands. Moreover, cyanide anions block the potential of Fe ion in heme to establish any coordinative interactions with other ligands. Our data showed that immuno-precipitation of metHb was abrogated in the presence of KCN. This result suggests that the co-precipitation of human IgG and metHb is dependent on heme and it is not due to pre-existing reactivity of antibodies to globin or non-specific binding of the protein to the beads with immobilized protein G.

Haptoglobin is a physiological Hb scavenger that mediates clearance of cell free Hb from the circulation. The interaction of haptoglobin with Hb is characterized by a very high affinity[54]. Binding of haptoglobin restricts the redox activity and cytotoxic activity of Hb. Here, we tested the effect of haptoglobin on the potential of human metHb to induce polyreactivity of IgG. Pre-incubation of metHb with haptoglobin resulted in a reduced but still considerable effect on the reactivity human IgG towards Factor IX and globin (Supplementary Fig. 3), thus suggesting that haptoglobin is not able to completely inhibit the effect of metHb on human antibodies.

Collectively, these data suggest that metHb and its close homolog myoglobin are capable of modifying the binding specificity of human IgG present in healthy antibody repertoires.

**The oxidation state of heme in Hb is important for induction of antibody polyreactivity and autoreactivity.** The oxidation state of iron in heme has important repercussions for the functions and integrity of Hb. The results presented above were obtained by using oxidized form of Hb, i.e., metHb. Interestingly, the transformation of metHb to reduced Hb ($Fe^{2+}$ heme) using the reducing agents, ascorbic acid or sodium dithionite, resulted in profound abrogation of the potential of the hemoprotein to impact the binding characteristics of human pooled IgG or of a monoclonal heme-sensitive IgG1 (Fig. 2a, b). Conversely, pre-exposure of metHb to hydrogen peroxide, which forms highly oxidized species of (ferryl-Hb), increased the potential of the hemoprotein to induced polyreactivity of human IgG (Fig. 2c). The data obtained with sodium dithionite and hydrogen peroxide, however, should be interpreted cautiously as these substances can induce post-translational effect on immunoglobulins

and hence exert some activities distinct from their impact of metHb.

Importantly, Hb is a redox-active protein that might oxidize other molecules by generation of ROS. The presence of catalase or superoxide dismutase in the reaction mixture did not inhibit the effect of metHb on antigen-binding specificity of pooled IgG (Supplementary Fig. 4a). Likewise, there was no potentiation in metHb-mediated induction of antibody polyreactivity when the reaction was performed in buffers prepared with heavy water ($D_2O$)—a condition that increases the lifetime of certain ROS (Supplementary Fig. 4b). These data ruled out the implication of bulk ROS liberation in metHb-induced polyreactivity of antibodies but they could not rule out the presence of minor localized oxidative changes induced by heme molecule when it interacted with IgG molecule.

Taken together, the results from this part of the study indicate that the oxidation state of heme is of essential importance for induction of polyreactivity and autoreactivity of human IgG. The oxidized heme species are considerably more potent in their effects on antibodies in comparison to the reduced ($Fe^{2+}$) form.

**Heme transfer from hemoproteins to IgG contributes to acquisition of polyreactivity and autoreactivity.** Because human IgG have been shown to gain reactivity to a large panel of antigens after incubation with hemoproteins, the above data suggest a central role of heme contained in Hb and myoglobin in conferring polyreactivity of vulnerable antibodies. Indeed, indispensable role of heme and its coordinated iron ion for effects of metHb was proven by the absence of any effect on human antibodies' immunoreactivity following an incubation with globin reconstituted with heme analog Zn(II) protoporphyrin IX or after an incubation of IgG with globin only (Supplementary Fig. 5). We thus investigated whether oxidized Hb uncovers polyreactivity of certain human antibodies via transfer of its heme moiety.

Such mechanism would explain the absence of any effect on antigen-binding properties of antibodies after exposure to cytochrome C (Fig. 1e)—a hemoprotein wherein heme moiety is covalently linked to the polypeptide chain and hence the heme transfer to antibodies could not occur. To test the hypothesis for heme transfer, we covalently conjugated myoglobin and

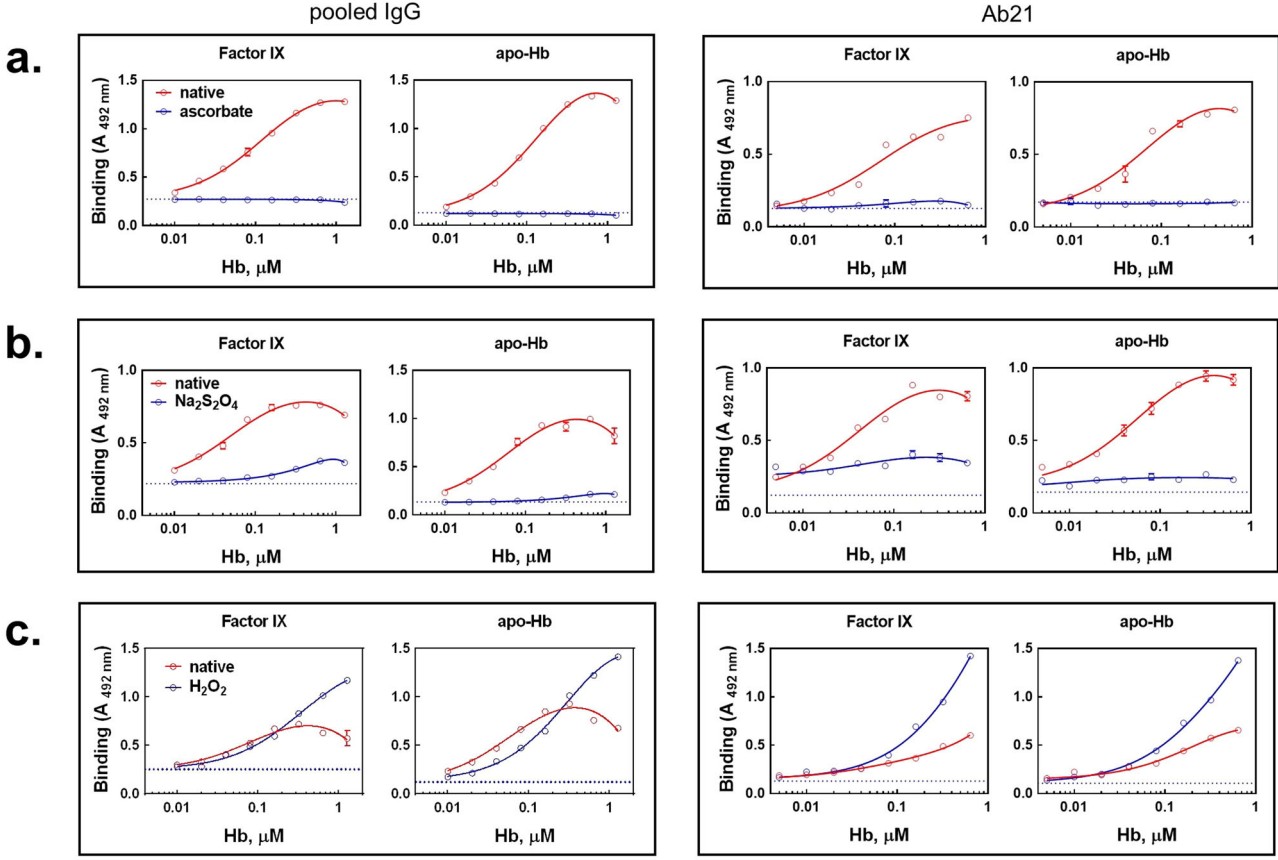

**Fig. 2 Pro-oxidative potential of heme plays an essential role in modification of antigen-binding specificity of antibodies by metHb.** Pooled human IgG and a monoclonal IgG1, Ab21 at 10 and 2 μM, respectively were exposed to increasing concentrations of metHb in the absence (red lines and symbols) or in the presence of: **a** 100 μM ascorbic acid; **b** 0.5- and 2-mM sodium dithionite, for pooled IgG and Ab21, respectively; and **c** 1 mM hydrogen peroxide (blue lines and symbols). After incubation for 30 min the samples were diluted 10× and incubated with immobilized factor IX and apo-Hb. Each data point represents mean absorbance ±SD from $n = 3$ repetitions of each sample. The dashed lines represent the reactivity of native IgG. Representative results from two independent experiments (pooled IgG) or one experiment (Ab21) are shown.

cytochrome C to Sepharose beads (Fig. 3a) Myoglobin was preferentially used in this experiment owing to its monomeric form (in contrast to Hb, which consists of four non-covalently bound protomers that can dissociate). Conjugation of the two hemoproteins to agarose beads allowed us to easily separate antibodies from the hemoproteins after the incubation. The obtained results indicated that exposure of a pooled human IgG preparation to Sepharose-bound myoglobin resulted in a transfer of heme to IgG molecules, as demonstrated by appearance of pseudo-peroxidase activity in flow-through containing IgG (Fig. 3b). In contrast, no pseudo-peroxidase activity was detected in the pooled IgG incubated with Sepharose-conjugated cytochrome c (Fig. 3b). To confirm that the transfer of heme was responsible for the induced polyreactivity of antibodies, we assessed the binding to human proteins of pooled IgG, which was pre-incubated with hemoproteins conjugated to agarose beads. Contact of human pooled IgG with beads with conjugated myoglobin, but not cytochrome c, resulted in a significant increase in the binding to Factor IX and apo-Hb (Fig. 3c).

Induction of antibody polyreactivity by heme transferred from metHb would require heme binding to a defined site on the IgG molecule. The interaction of protoporphyrin compounds devoid of iron ions to heme-sensitive antibodies, such as the one used in this study, i.e., Ab21, does not result in appearance of novel antigen specificities[59]. We thus exploited hematoporphyrin IX, an iron ion-devoid analog of heme, to further substantiate the key role of the heme transfer (Fig. 3d). Pooled and monoclonal IgG

(Ab21) were first exposed to hematoporphyrin IX, and then incubated with increasing concentrations of metHb. As depicted on Fig. 3e, saturation of the heme binding sites on IgG by hematoporphyrin IX resulted in a substantial reduction in the potential of metHb to induce reactivity to Factor IX and apo-Hb.

To provide further evidence that the transfer of heme from metHb to IgG molecule is responsible for modification of antigen binding specificity, we investigated the effect of a plasma heme scavenging protein—hemopexin[54]. Our rationale to use hemopexin was based on the fact that this protein binds heme with enormous affinity ($K_D \sim 10^{-14}$ M), and hence can intercept the released heme from metHb, thus preventing the contact of heme with IgG. The incubation of heme-sensitive Ab21 with increasing concentrations of metHb (0.005–0.64 μM, corresponding to total heme concentrations in the range of 0.02–2.56 μM) in the presence of a fixed concentration of hemopexin (0.32 μM) resulted in a pronounced reduction of antibody reactivity towards human proteins as compared to the antibody incubated with metHb only (Fig. 4a). To rule out the possibility that hemopexin serves as autoantigen for Ab21, and thus it blocked the metHb-induced polyreactivity, we incubated Ab21 with metHb in the presence of apo-hemopexin or in the presence of hemopexin that was pre-loaded with heme. The heme-hemopexin (1:1) complex demonstrated a substantially reduced capacity to block the effect of metHb on Ab21 (Fig. 4b).

Our data demonstrated that blocking heme capacity to establish coordinative interactions by excess of cyanide anions

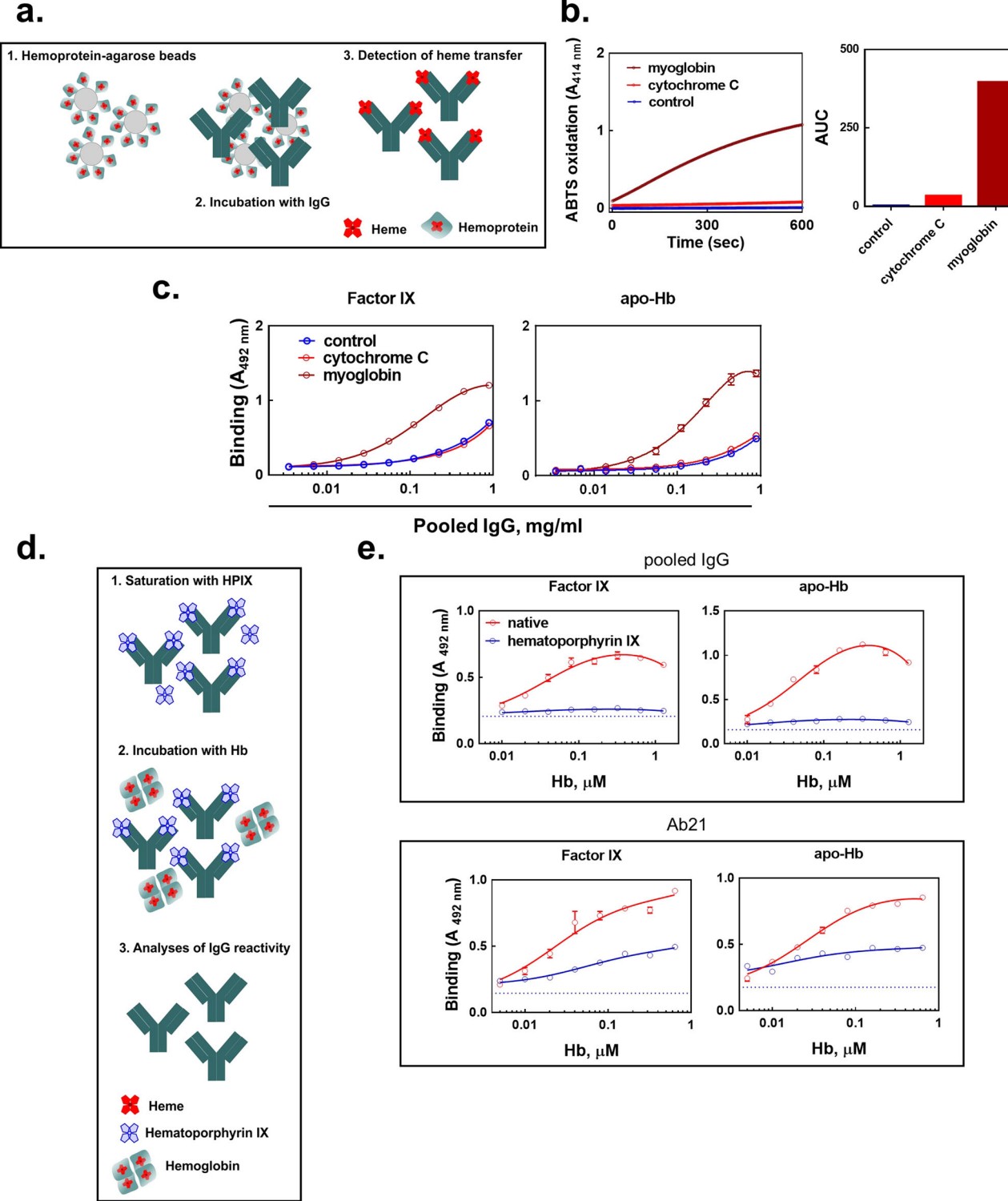

results in a marked drop in the potential of metHb to modulate the antigen-binding functions of human antibodies (Fig. 4c and Supplementary Fig. 6). To better characterize the transfer of heme from metHb and the effects of hemopexin and cyanide anions, we applied absorbance spectroscopy analyses. The data for time resolved spectral changes in the UV-vis absorbance range (Fig. 4d), confirmed the dissociation of heme from metHb. The transfer of heme from metHb to hemopexin was clearly observed by a red shift in the absorbance maximum at Soret region (from 405 to 410 nm) and the characteristic spectral change in the low energy region of the spectrum (Fig. 4d). Interestingly, the spectroscopy analyses indicated that the considerable changes in absorbance spectra occurred in the first 30 min after mixing of the proteins. These data also demonstrated that the time-dependent spectral changes were inhibited in the presence of cyanide anions or reduced in the case when hemopexins was pre-loaded with heme. When a hemoprotein contains covalently bound heme, like cytochrome, c the transfer of heme towards hemopexin was also not detected (Fig. 4d). This observation proved the presence of heme dissociation from metHb and could explain the observed

**Fig. 3 Transfer of heme from hemoproteins to human pooled IgG. a** Scheme of the experiment for assessment of transfer of heme from hemoproteins to IgG. **b** Determination of pseudo-peroxidase activity of heme in human IgG. Pooled IgG (20 µM) was incubated for 3 h at room temperature with Sepharose-conjugated myoglobin or cytochrome c. The presence of heme in IgG-containing flow through was assessed by kinetics of oxidation of ABTS (kinetic curves and their corresponding areas under the curves). Blue circles, line and bars depict pseudo-peroxidase activity of native IgG; red circles, line and bar depict pseudo-peroxidase activity of IgG incubated with Sepharose-bound cytochrome c; brown circles, line and bar depict pseudo-peroxidase activity of IgG incubated with Sepharose-bound myoglobin. **c** Immunoreactivity of IgG incubated with Sepharose-conjugated hemoproteins. Native IgG (blue line and symbols); IgG pre-incubated with Sepharose-bound cytochrome c (red line and symbols) or Sepharose-bound myoglobin (brown line and symbols) were incubated at increasing concentrations (0; 0.00347–0.89 mg/ml) with immobilized factor IX and apo-Hb. Each point represents mean absorbance ±SD from $n = 2$ repetitions of each sample. **d** Scheme of the experiment for assessment of the capacity of hematoporphyrin IX to block metHb-mediated induction of polyreactivity of IgG. **e** Inhibitory effect of hematoporphyrin IX on metHb-induced autoreactivity of IgG. Pooled human IgG at 10 µM or monoclonal IgG1 (Ab21) at 2 µM were first exposed to hematoporphyrin IX at 20 and 10 µM, respectively. After native IgGs (red lines and symbols) or hematopophyrin IX exposed IgGs (blue lines and symbols) were treated with increasing concentrations of metHb. The binding of antibodies to immobilized factor IX and apo-Hb was evaluated by ELISA. Each data point represents mean absorbance ±SD from $n = 3$ repetitions of each sample.

inhibitory effects of hemopexin and cyanide anions in the immunoassays (Fig. 4a–c) as well the absence of effect of cytochrome c on the antigen binding reactivity of human IgG (Fig. 1e).

Collectively the results from different complementary assays suggest that the transfer of heme from hemoproteins (metHb and myoglobin) to IgG molecules is indispensable for induction of antibody polyreactivity and autoreactivity.

**Interaction of heme with IgG is indispensable for induction of polyreactivity and autoreactivity after transfer from metHb.** Ab21 is a well-characterized human IgG1 antibody that acquire polyreactivity upon contact with heme[59]. In a previous study, we modeled the interaction of heme with variable region of Ab21, using computational tools[60]. This model revealed that the most probable binding site for heme overlaps with the antigen binding site of the antibody[60] (Fig. 5a). More specifically, the result suggested that heme predominantly establishes contacts with CDR3 loop of heavy chain (Fig. 5a). Inspection of the sequence of CDR H3 of Ab21 (Fig. 5b) revealed that this region is rich of amino acid residues that can directly interact with heme. Thus, CDR H3 of Ab21 contains four positively charged amino acid residues (Lys and Arg) that can establish ionic bonds with the two negatively charged propionate residues of heme. Furthermore, the CDR H3 contains a stretch of four Tyr residues. Tyr is known to interact with heme both via aromatic and metal coordination interactions and this residue is often enriched in heme-binding sites of proteins[61]. To provide evidence that release of heme from metHb result in binding of heme to IgG molecule and this binding is responsible for acquisition of polyreactivity and autoreactivity, we performed site-directed mutagenesis analyses of Ab21. We mutated residues in CDR H3 that potentially can interact with heme. The mutants are indicated on Fig. 5b. Moreover, to understand the role of somatic mutations for heme-induced changes in antigen-binding specificity of antibodies, we reverted by site-directed mutagenesis the sequences of variable regions of heavy and light chains of Ab21 to its germline configuration. All mutants of Ab21 were successfully expressed. The effect of exposure of the mutants of Ab21 to metHb was evaluated by ELISA. The obtained results indicated that most of the single mutants in CDR H3 of Ab21 retain sensitivity to metHb, although the effect of metHb was reduced as compared to the one observed for unmutated Ab21 (Fig. 5b). Nevertheless, a single mutant Y111A or replacement of all Tyr residues caused an almost complete abrogation of capacity of metHb to affect specificity of Ab21 (Fig. 5c). These results suggested that the presence in CDR H3 region of Ab21 of amino acid residues that are able to bind heme, especially Tyr, is of critical importance for induction of polyreactivity of the antibody. Furthermore, reversion of Ab21 to its germline progenitor variant resulted in a considerable

reduction (>3 folds) of the sensitivity of the antibody to metHb (Fig. 5c).

Collectively, these data suggest that the unique sequence configuration of variable region of Ab21 is indispensable for induction of polyreactivity and autoreactivity following exposure to metHb. We also identified the residues of critical importance for heme binding and induction of polyreactivity. Moreover, we demonstrated that the sensitivity to heme is a feature optimized during somatic hypermutation of the antibody.

**Quantitative aspects of interaction of heme-induced polyreactive IgG with hemoproteins.** Our data indicate that metHb induces a potent reactivity of human antibodies to its very protein scaffold – globin. Recognition of cell free metHb by endogenous antibodies under hemolytic conditions may have important physiological consequences. To provide an understanding about the mechanism of interactions of hemoproteins with heme-sensitive antibodies, we performed kinetic and thermodynamic analyses. For these analyses we selected Ab21, as it demonstrated high sensitivity to heme. The antibody was subjected to real-time binding analyses by using a surface plasmon resonance-based biosensor assay. Kinetic analyses indicated that heme-exposure results in a relatively strong binding of Ab21 to both Hb and myoglobin ($K_D$ values in the range 9-75 nM) (Fig. 6a and Supplementary Data 1), suggesting that such interactions may be of relevance under physiological conditions. Further, the kinetic rate constants and equilibrium affinity of the interactions were determined as a function of temperature (Supplementary Data 1). Arrhenius plots depicted on Fig. 6b indicated that changes in temperature had different effects on the binding kinetics of the heme-complexed Ab21 to the different hemoproteins. Thus, the association rate of Ab21 was not sensitive to the temperature in the case of binding to metHb, while it was very sensitive to temperature changes in the case of binding to myoglobin (Fig. 6b).

The dissociation rate constant of Ab21 was considerably temperature sensitive for both myoglobin and metHb, but demonstrated opposite effects for the two hemoproteins (Fig. 6b). Thus, values of $k_d$ were elevated in case of metHb and diminished that of myoglobin, as the reaction temperature increased. Protein-dependent differences in the temperature sensitivity of the interactions were also observed in the case of the equilibrium affinity ($K_D$ values) (Fig. 6b). Distinct impacts of the temperature on the binding kinetics and affinity of a given heme-sensitive antibody imply that the recognition of metHb and myoglobin occurs by different mechanisms. Indeed, thermodynamic analyses revealed quantitative and qualitative differences in enthalpy and entropy changes for interactions of Ab21 with metHb and myoglobin (Fig. 6c). For example, the equilibrium thermodynamic parameters indicated that binding of this antibody to myoglobin is entropy-driven and enthalpy-controlled process

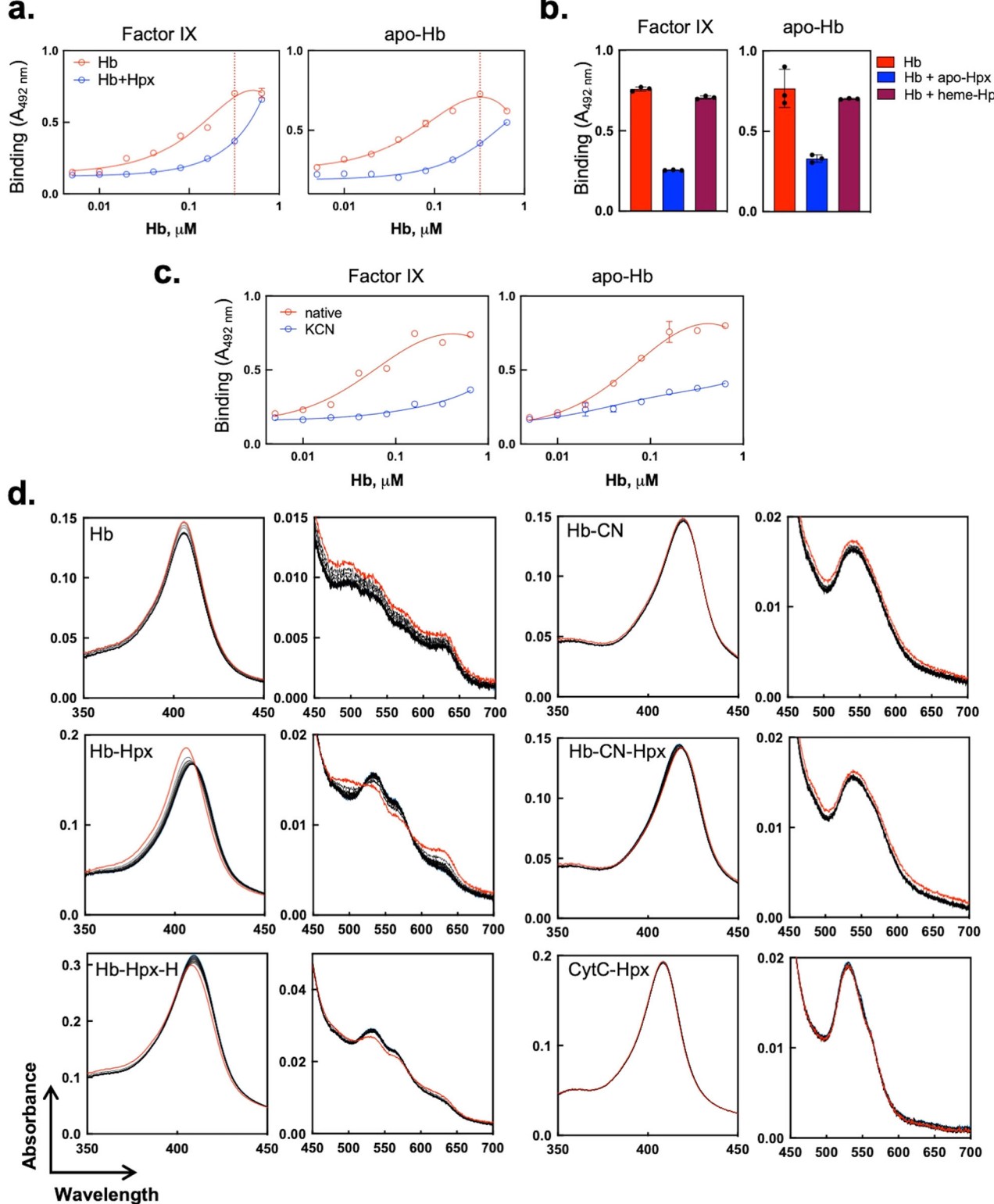

(positive values for both TΔS and ΔH). In contrast, the binding of heme-exposed Ab21 to metHb is driven both from favorable changes in the equilibrium enthalpy and entropy (positive value of TΔS and negative value of ΔH) (Fig. 6c). Kinetic and thermodynamic data (e.g., favorable effect of temperature on association rate constant and a negligible change in the value of association TΔS) suggest that the recognition of myoglobin is not accompanied by significant conformational changes in the interacting proteins. In contrast, the parameters of Ab21 binding to metHb (e.g., absence of temperature sensitivity of association rate and more pronounced changes association TΔS) suggest the occurrence of conformational changes in the interacting proteins.

Collectively, these results indicate that recognition of hemoproteins by heme-sensitive antibodies are characterized by physiologically relevant binding affinities and by protein-dependent differences in the binding mechanisms.

**Fig. 4 Transfer of heme from metHb to human IgG—inhibition by hemopexin and KCN. a** Immunoreactivity of Ab21 pre-incubated at 2 μM with increasing concentrations of metHb (0, 0.005–0.64 μM) in the absence or in the presence of 0.32 μM human hemopexin (Hpx). The concentration of Hpx as compared with metHb concentration was indicated by vertical dashed lines on the graphs. Each data point represents mean absorbance ±SD from $n = 3$ repetitions of each sample. A representative result from two independent experiments is shown. **b** Immunoreactivity of Ab21 pre-incubated at 2 μM with 0.32 μM metHb in the absence (red bars) or the presence of 0.5 μM apo-hemopexin (Hpx, blue bars) or 0.5 μM holo-hemopexin (maroon bars). The holo-hemopexin was prepared by incubation of 50 μM hemopexin with 50 μM hematin for 5 min prior dilution. Each bar represents mean absorbance ±SD from $n = 3$ repetitions of each sample. The individual values are presented in black circles. The Kruskal–Wallis statistical test indicated significant difference between the conditions with p values equal to 0.05 and 0.004 for apoHb and FIX, respectively. **c** Ab21 at 2 μM was pre-treated with increasing concentrations of metHb (0, 0.005–0.64 μM) in the absence or in the presence of 2 mM KCN. Each data point represents mean absorbance ±SD from $n = 3$ repetitions of each sample. **d** Absorbance spectra of metHb (0.5 μM) alone or in the presence of: (1) 2 μM apo-Hpx; (2) 2 μM holo-Hpx (Hpx-H); (3) 2 mM KCN; (4) 2 mM KCN and 2 μM apo-Hpx. The spectra of 2 μM cytochrome c in the presence of 2 μM of apo-Hpx is also shown. The spectra in high energy region (350–450 nm) and low energy region (450–700 nm) were shown with different scales of y-axes. The spectra recorded immediately after mixing of the proteins is displayed in red thick line. The spectra recorded at last time point is presented in blue thick line but clusters with other spectral readings, and it is barely visible. Of note cytochrome c was diluted to 2 μM to match heme amount introduced with 0.5 μM metHb. The spectra were taken at intervals of 30 min for total time of 8 h.

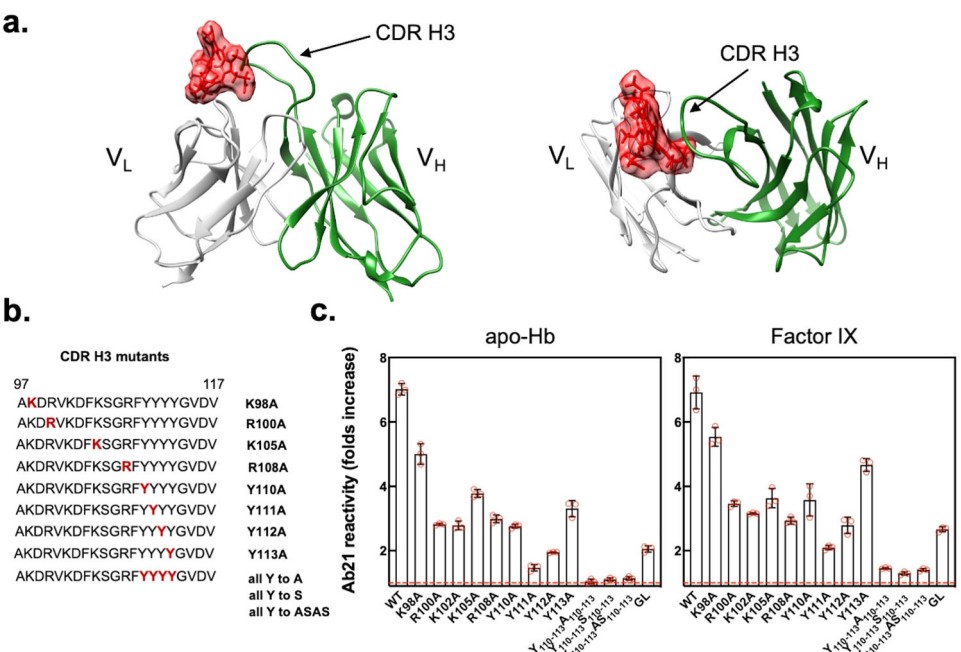

**Fig. 5 Molecular features of antigen binding site of a monoclonal IgG acquiring polyreactivity and autoreactivity after exposure to metHb. a** Structural model of the complex of variable region of Ab21 and heme. The picture was generated using data from structural modeling of Ab21 variable region and molecular docking that were already published in ref. [60]. Light chain variable domain is depicted in gray, heavy chain variable domain is depicted in green. Protoporphyrin IX ring (used for the docking) was depicted in red. Side and top views of the variable region are shown. The models were built using UCSF Chimera software v. 1.16. **b** A plot depicting the sequence of CDR H3 of Ab21 and the residues that were mutated (highlighted in red). The individual mutants are listed on right. **c** Immunoreactivity of Ab21 mutants towards human factor IX and human apo-Hb. Ab21 wt, its mutants and its germline variant (GL) were diluted to 1 μM and incubated with 0.16 μM metHb. After, all samples were diluted 10 folds and incubated with the immobilized antigens. The plots depict the ratio of binding of each antibody variant after treatment with metHb versus the binding of the Ab21 in the absence of treatment (background reactivity). The red dashed line indicates the ratio of 1 corresponding to absence of effect of metHb. Each bar represents mean ratio ±SD from $n = 3$ repetitions of each sample. The individual values are presented in red circles.

## Discussion

Here we show that human antibodies acquire polyreactivity and autoreactivity after incubation with sub-equimolar concentrations of oxidized hemoproteins, such as hemoglobin or myoglobin. The changes in the reactivity of human antibodies were mediated by a heme transfer from hemoprotein to IgG molecule. This phenomenon might have repercussions for understanding the physiopathology of hemolytic conditions.

Numerous studies have reported that patients with distinct hemolytic diseases that lack immune etiology (SCD, malaria) often have high titers of autoantibodies in their plasma as well as glomerular deposits of immunoglobulins[62–68]. Antigen binding profiles of these autoantibodies, resembles autoantibodies detected in patients with systemic autoimmune diseases, i.e., they often recognize ribonucleoproteins, DNA, and phospholipids[68–73]. The origin of the autoantibodies in hemolytic diseases and their contribution to the pathology remains unknown. A plausible explanation for the generation of the autoantibodies is an impairment of the immune tolerance due to chronic inflammation and/or exposure to large amounts of intracellular proteins. However, results obtained in the present study suggest another explanation for appearance of autoantibody reactivities under hemolytic conditions.

Besides recognition of autoantigens, we also observed that metHb induced a reactivity of antibodies towards its protein component – globin. Importantly, addition of metHb to normal human serum

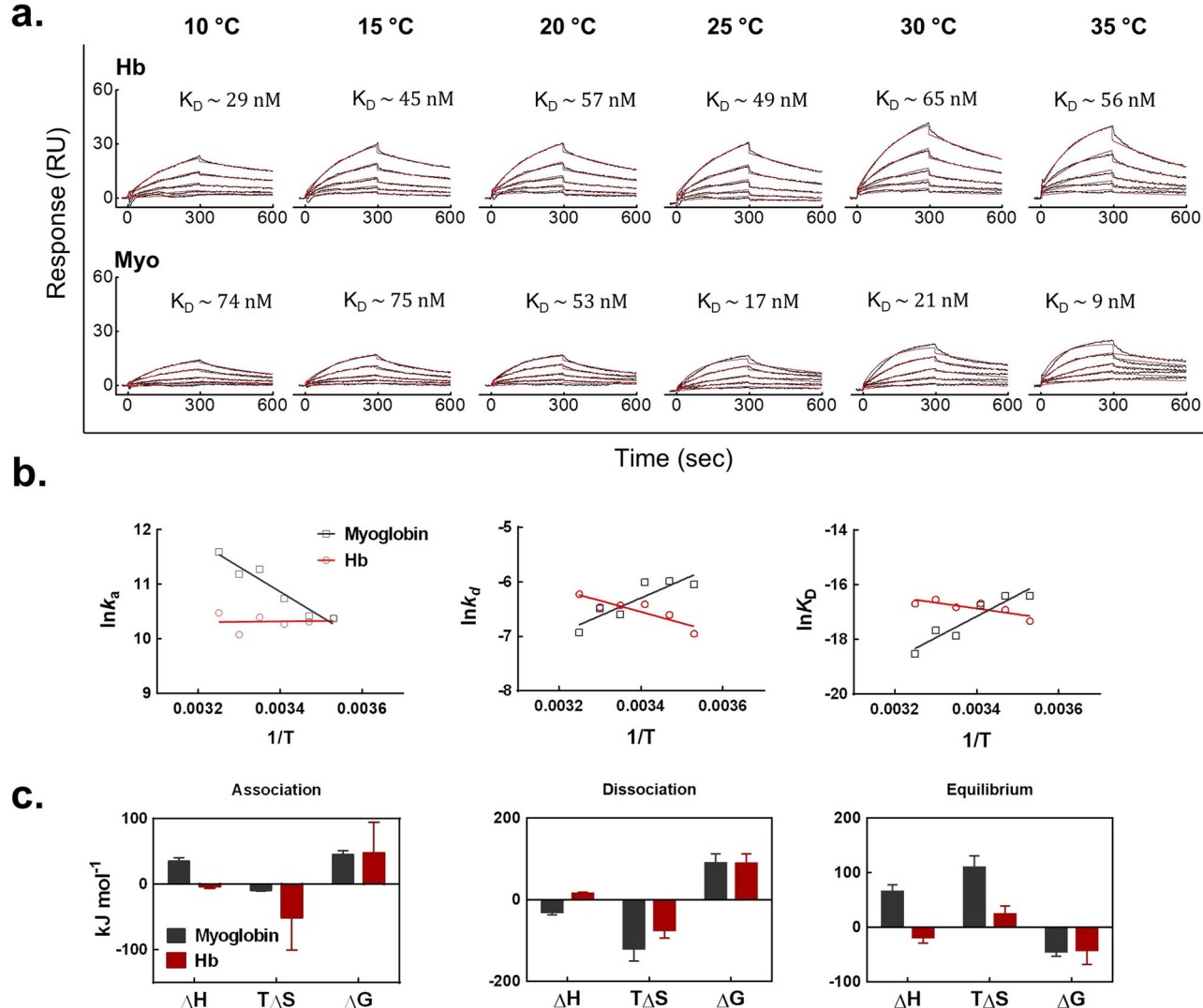

**Fig. 6 Quantitative analyses of interaction of Ab21 with Hb and myoglobin. a** Real-time interaction profiles of heme-exposed Abs21 to immobilized on sensor surface metHb and myoglobin. The black line represents the binding profiles obtained after injection of serial dilutions of Ab21 (125–7.81 nM). The red lines depict the computed fits of data obtained by global kinetic analyses using BIAevaluation software. The kinetic measurements were performed as a function of temperature. Representative sensorgrams from two independent experiments are shown. The average values of equilibrium dissociation constant obtained by three independent fittings of the experimental data are shown as inserts. A table with all kinetic values, applied fitting models and statistics is presented as Supplementary Data 1. **b** Arrhenius and Van't Hoff plots depicting the temperature dependency of kinetic rate constant and equilibrium dissociation constant of binding of Ab21 to metHb (red line and symbols) and myoglobin (black liens and symbols). The plots were built by linear regression analyses using Graph Pad Prism software v. 6.01. **c** Changes in activation and equilibrium thermodynamic parameters of interaction of Ab21 to metHb (red bars) and myoglobin (gray bars). The values of the thermodynamic parameters were calculated by Eyring's analyses using slopes of Arrhenius plots depicted in **b**. Each bar shows change of binding energy ±SE, reflecting the standard errors from the linear regression fits on **b**.

resulted in formation of immune complexes with endogenous antibodies, an observation suggesting that this phenomenon might occur under hemolytic conditions in vivo. The biological significance of post-translational acquisition of reactivity of IgG to metHb remains to be estimated. One of the most prominent pathogenic effects of extracellular metHb and myoglobin is their potential to cause kidney failure. One can speculate that the formation of complexes with IgG reduces the toxicity of the hemoproteins, as the increase in the overall molecular weight will prevent renal filtration and a subsequent tubular injury. This mechanism would be of importance in situations when the high affinity scavenger of Hb (haptoglobin) is overwhelmed. The formation of complexes with endogenous IgG can facilitate clearance of hemoglobin and red blood cell debris through phagocytes via Fc-γ receptors.

In a previous report it was shown that IgG antibodies from healthy individuals can recognize Hb[74]. Significant binding of antibodies was observed, however only when Hb is bound to soluble CD163. The formation of the ternary complex was proposed to contribute to the clearance of Hb via binding to Fc-γ receptors on macrophages. Another study has demonstrated that Hb is among the most prominent targets of the natural IgM autoantibodies in newborn humans[75]. It is noteworthy that in systemic lupus erythematous and some hemolytic diseases a presence of autoantibodies specific for Hb was also documented[76]. It remains, however, unknown whether there is a link between these reports and the phenomenon observed in our study. But in all cases, it appears that Hb is a preferred target of antibodies.

Mechanistic studies revealed that the potential of oxidized Hb or myoglobin to modify the specificity of human IgG depends on a transfer of heme from the hemoprotein to IgG. Previous studies have demonstrated that the exposure of human antibodies to heme results in an appearance of novel specificities to self- and pathogen-derived antigens[40–45,47,77,78]. Two alternative mechanisms were proposed to explain heme's effect on immunoglobulins. The first suggest that the pro-oxidative potential of heme plays an essential role. A convincing argument supporting this hypothesis is that other pro-oxidative substances such as iron ions and ROS are also capable (although less efficiently than heme) to induce novel antigen-binding specificities of antibodies[41,79,80]. The second hypothesis proposes that heme binds to antigen binding sites of the immunoglobulins and serves as interfacial bridge for interaction with other proteins[60]. Indeed, heme molecule is characterized with a broad binding promiscuity to human proteins[81,82] Therefore, once heme is bound in an appropriate steric orientation to an antibody it might endow the latter with antigen binding polyreactivity. Evidences for this mechanism have been obtained by spectroscopy, antigen-binding assays and molecular modeling[43,45,59,60]. We hypothesized that effect of metHb is due to a direct transfer of heme from hemoproteins to immunoglobulins. To prove this hypothesis, we performed series of experiments (results presented on Figs. 3–5). The data from these experiments confirmed the dissociation of heme from metHb and myoglobin (but not from cytochrome C, where heme is bound covalently). These data also proved the direct binding of hemoprotein-derived heme to IgG molecules. Previous studies have shown that oxidized form of heme ($Fe^{3+}$) has lower affinity for polypeptide chain of hemoprotein molecule (Hb or myoglobin) as compared to the reduced one ($Fe^{2+}$)[83,84]. This fact can explain our data showing that in the presence of reducing agents (ascorbic acid, dithionite) or agents that block metal coordination interactions of heme (cyanide anions) there was a dramatic reduction of the effect of metHb on human antibodies. The presence of heme scavenging protein hemopexin also inhibited the effect of hemoproteins on antibodies—another demonstration that free heme had an important role for the induction of changes in the antigen binding potential of certain human Abs.

A further support for heme transfer from metHb to IgG was obtained by using hematoporphyrin IX, a heme analog that has a similar structure to heme and is devoid of iron ion. Presumably, this compound could compete with heme for binding to the same sites on IgG molecule but could not induce polyreactivity, as the presence of iron in porphyrin molecule is indispensable for effect of heme on antibodies[59]. Our data revealed that the pre-incubation of IgG with hematoporphyrin IX considerably reduced the ability of metHb to trigger polyreactivity and autoreactivity of the studied antibody. These data suggest that the direct heme binding to antibody is indispensable for acquisition of polyreactivity and autoreactivity. Further to characterize the features of the binding site for heme on human IgG we performed site-directed mutagenesis analyses with a model heme-sensitive antibody, Ab21. These analyses identified residues in CDR H3 region of the antibody that has a major contribution for acquisition of polyreactivity upon contact with metHb. Thus, replacement of certain residues in the most diverse and important for antigen binding specificity region of antibodies, i.e., CDR3 of heavy chain, were found to have profound effect on the capacity of metHb to modify the specificity of our model monoclonal human IgG1. More specifically, mutation of a single Tyr residue (Y111A) or a stretch of four Tyr residues in CDR H3 resulted in an almost complete loss of sensitivity of the antibody to the effect of metHb (Fig. 5). Importantly, Tyr residues are known to establish direct interactions with heme molecule and is one of the residues enriched in the binding sites for heme on diverse

proteins[61,85,86]. These results corroborate our previous data from molecular modeling studies of the same antibody, indicating that heme-binding site on Ab21 overlaps mostly with CDR H3[60]. Moreover, the site-directed mutagenesis analyses implied that heme in order to induce polyreactivity should be transferred from hemoprotein and bind to a defined and specific site overlapping with the paratope of the antibody. The results from mutagenesis also highlight that the unique configuration of antigen-binding site of the antibody optimized during affinity maturation process is a pre-requisite for the ability of hemoproteins to induce polyreactivity, thus suggesting that this phenomenon is highly specific.

We performed kinetic and thermodynamic analyses to gain insights in the polyreactive recognition of hemoproteins by heme-induced antibody. These analyses demonstrated quantitative and qualitative differences in the mechanism of binding of the heme-bound antibody to metHb and myoglobin. The differences in the changes of activation and equilibrium enthalpy strongly suggest contribution of different types of non-covalent forces in the formation of complexes of antibody with these hemoproteins. This can be explained by involvement of different amino acid residues in the interactions of metHb or myoglobin with the heme-bound Ab21. Despite these differences in the binding energetics the heme-sensitive monoclonal IgG1 recognized both proteins with relatively high binding affinity ($K_D$ values in low nanonmolar range). These data suggest that formation of complexes of heme-sensitive antibodies with hemoproteins might have a pathophysiological relevance.

The natural polyreactive antibodies are important component of the immune repertoires[87–89]. They were shown to contribute for immune homeostasis by clearance in tolerogenic manner of damaged macromolecules or dead cells[90–92]. Here we demonstrate that a pro-inflammatory and pro-oxidative hemoproteins hemoglobin and myoglobin, released in massive amounts during hemolysis or muscle damage, can trigger polyreactivity and autoreactivity of some human IgG antibodies. Our data highlighted that induction of polyreactivity potential and reactivity to protein component of Hb occurs via a transfer of the heme. This phenomenon can have beneficial consequences facilitating the removal of the toxic proteins from circulation.

These findings can serve as a basis for further investigations for understanding of origin of autoantibodies in hemolytic disease and for better understanding of physiopathology of hemolytic diseases in general.

## Methods

**Hemoproteins and antibodies.** Human and bovine oxidized Hb (or methemoglobin, metHb) equine myoglobin and equine cytochrome c were all obtained in lyophilized form (Sigma-Aldrich, St. Louis, MO, USA). Fresh stock solutions of hemoproteins in PBS were prepared immediately before use and kept at ice in dark. Apo-Hb was prepared from human Hb by using a procedure of heme extraction by acidified acetone as described in[93]. Therapeutic intravenous immunoglobulin (IVIg, Endobulin, Baxter, USA) that consists of pooled human IgG purified from >3000 blood donors, was used as a source of polyclonal IgG from healthy individuals. A monoclonal IgG1 antibody—Ab21 that was previously identified as able to acquire polyreactivity upon interaction with heme—was used as a representative heme-sensitive antibody throughout the study[59]. Sera obtained from anonymous healthy blood donors (under the ethical convention between INSERM with Etablissement Français du Sang—15/EFS/012) were used as a source of normal human IgG.

**Enzyme-linked immunosorbent assay (ELISA)**
*Preparation of samples.* For treatment of antibodies with metHb, human pooled IgG preparation was diluted to 10 μM (1.5 mg/ml) in PBS and exposed to following concentrations of metHb (or other studied hemoproteins)—0, 0.01, 0.02, 0.04, 0.08, 0.16, 0.32, 0.64, and 1.28 μM (including 2.56 and 5.12 μM of metHb, only in case of ELISA results presented on Fig. 1a or when myoglobin and cytochrome c were used for treatment). Typically, Ab21 was diluted to 2 μM (0.3 mg/ml) and incubated in the presence of 0, 0.005, 0.01, 0.02, 0.04, 0.08, 0.16, 0.32, and 0.64 μM of Hb. The incubation of human immunoglobulins with metHb was performed for 30 min at

ice. The treatment conditions were selected as based on preliminary optimization experiments. In some experimental settings, the reaction mixture contained one of the following proteins or substances: 10 IU/ml of bovine catalase (Sigma-Aldrich); 10 IU/ml bovine superoxide dismutase (Sigma-Aldrich); 0.5 μM human haptoglobin (Sigma-Aldrich); 0.32 μM human hemopexin (Sigma-Aldrich), or 0.5 μM when the effect of holo-hemopexin was studies; 1 mM $H_2O_2$ (Merck, Darmstadt, Germany); 2 mM KCN (Sigma-Aldrich); 0.5- or 2-mM sodium dithionite ($Na_2S_2O_4$, Sigma-Aldrich) or 100 μM ascorbic acid (Sigma-Aldrich). The indicated proteins or substances were added to immunoglobulin solutions prior addition of metHb.

In a particular case, the reaction buffer (PBS) for treatment of immunoglobulins with metHb was prepared by using deuterium oxide (Sigma-Aldrich) as a solvent. In another experimental setting 10 μM of pooled IgG or 2 μM of monoclonal IgG1 were pre-incubated for 15 min on ice in the presence of 20 or 10 μM, respectively, of hematoporphyrin IX (Sigma-Aldrich). After antibodies were incubated with metHb as stated above.

*Parameters of a typical ELISA procedure.* In a typical experimental setting, 96-well microtiter polystyrene plates (Nunc Maxisorp, Roskilde, Denmark) were coated with 10 μg/ml of human factor IX (LFB, Les Ulis, France) or human apoHb. The proteins were diluted in PBS and incubated with the microtiter plates for 2 h at room temperature. After, the residual binding sites on plates were blocked by PBS containing 0.25% Tween 20 (Sigma-Aldrich). Pooled IgG and monoclonal IgG1 (exposed to metHb in the absence or presence of different substances) were diluted in PBS containing 0.05% Tween 20 (T-PBS) to 1 and 0.2 μM, respectively, and incubated with immobilized proteins for 2 h at room temperature. After extensive washing with T-PBS plates were incubated with mouse anti-human IgG conjugated with HRP (clone JDC-10, Southern Biotech, Birmingham, AL, USA). The immunoreactivity was revealed after extensive washing with T-PBS and addition of substrate solution, o-phenylenediamine dihydrochloride (SIGMAFAST™, Sigma-Aldrich). The reaction was stopped by 2 M HCl and the absorbance of 492 nm was recorded by using TECAN Infinite® 200 PRO microplate reader.

*Parameters of a polyreactivity ELISA.* In the case where the polyreactivity of human monoclonal IgG1 was assessed, the microtiter plates were coated with a panel of antigens: human factor IX (LFB); human factor VIII (Kogenate FS, Bayer HealthCare); porcine tubulin (Sigma-Aldrich), human C-reactive proteins (Calbiochem, San Diego, CA, USA), human factor B (CompTech, Tyler, TX, USA) human hemoglobin (Sigma-Aldrich); human VWF (Wilfactine, LFB), bovine myelin basic protein (Sigma-Aldrich); porcine thyroglobulin (Sigma-Aldrich); diphtheria toxoid and lipopolysaccharides from *Escherichia coli* O55:B5 (Sigma-Aldrich). This panel was selected to represents diverse and unrelated autoantigens and foreign antigens. Ab21 at 5 μM was exposed to 1 μM of metHb for 1 hour at ice. After the antibody was diluted 10× in T-PBS and incubated with the immobilized proteins for 2 h at room temperature. Next steps of the experiment are identical as those described above.

**Immunoblot analyses.** A total lysate of human umbilical vein endothelial cells (HUVEC) was mixed with an aliquot reducing sample buffer resulting in a final protein concentration of 1.5 mg/ml. Single well SDS-PAGE gels (NuPAGE Bis-Tris 10% gels, Novex, Life Technologies, Thermo Fisher Scientific, Waltham, MA, USA) were loaded with 75 μg of HUVEC proteins per gel. The proteins were separated at constant voltage of 200 V using Bio-Rad Mini-PROTEAN system (Bio-Rad, Hercules, CA, USA). After the electrophoretic separation, proteins were transferred on nitrocellulose membranes by using Mini Trans-Blot Cell electrotransfer system Bio-Rad. Nitrocellulose membranes were blocked overnight in TBS, containing 0.1% Tween 20 (T-TBS) at 4 °C. Human pooled IgG (Endobulin, Baxter) at 10 μM (1.5 mg/ml) was exposed to metHb at 0, 0.01–2.56 μM (twofold dilution step) concentrations in PBS and incubated for 30 min on ice. Ab21 was diluted to 1 μM (150 μg/ml) and incubated in PBS in the presence of 0, 0.025, 0.05, 0.125, 0.25, 0.5–1 μM of metHb. A normal human serum was first diluted 2× in PBS and then exposed to 50 μM of metHb. After the immunoglobulins were diluted in T-TBS to 1 μM (150 μg/ml) and 0.5 μM (75 μg/ml) for pooled IgG and monoclonal antibody, respectively; the serum was diluted 100×, 200×, and 400× in T-TBS. All samples were then incubated with nitrocellulose membranes for 2 h at room temperature by using miniblot system (Immunetics, Cambridge, MA, USA). After the incubation, the nitrocellulose membranes were washed for 1 h with T-TBS with six consequent changes of washing buffer (T-TBS). Next the membranes were incubated for 1 h at room temperature with a goat anti-human IgG, conjugated with alkaline phosphatase (Southern Biotech), diluted 3000× in T-TBS. After thorough washing with T-TBS for 1 h, immunoreactivities of different samples of IgG was revealed by addition of BCIP/NBT (Sigma-Aldrich) substrate solution.

**Absorbance spectroscopy.** UV-vis absorption spectra were recorded by using Agilent Cary-300 spectrophotometer (Agilent Technologies, Santa-Clara, CA). The measurements were performed at room temperature in quartz optical cells (Hellma, Jena, Germany) with 1 cm optical path. All samples were diluted in PBS to a final volume of 1 ml. The spectra of 0.5 μM of metHb alone or of 0.5 μM metHb in the presence of: (1) 2 μM hemopexin; (2) 2 mM KCN; (3) 2 mM KCN and 2 μM hemopexin, and (4) 2 μM holo-hemopexin (prepared by pre-incubation of 50 μM

hemopexin with 50 μM hematin for 5 min) were recorded in wavelength range 350–700 nm with data acquisition speed 300 nm/min. The spectra of cytochrome c diluted at 2 μM in the presence of 2 μM hemopexin were recorded at the same conditions. For observing the time-dependent spectral changes, the UV-vis spectra were measured immediately after diluting and/or mixing of the proteins and at 30 min intervals for total time of 8 h (in total 16 measurements for each condition). As the absorbance spectra of heme bound to Ab21 (published in ref. [59]), overlaps with the absorbance spectra with oxidized Hb (both have absorbance maximum at 405 nm in Soret region) we were not able to determine the transfer of heme from metHb to this antibody by absorbance spectroscopy.

**Site-directed mutagenesis of Ab21.** Identification of amino acids to be mutated was done after sequence alignment between mAb21 and closest germline sequence thanks to IMGT/V-QUEST tool. Wild-type codon was substituted by the mutagenic codon using In-Fusion technology (Takara) according to the manufacturer's protocol. Plasmid encoding wild-type sequence was linearized by PCR using overlapping primers containing the mutagenic codon. Plasmid was circularized using In-Fusion HD Enzyme (Takara), as per manufacturer's instruction and used to transform *E. coli* Stellar competent cells (Takara). Transformed bacteria were selected on LB plates containing 100 μg/mL ampicillin and plasmidic DNA was extracted with QIAprep Spin Miniprep Kit (Qiagen). Proper insertion of the desired mutation was controlled by sequencing the plasmid (Sanger sequencing, Eurofins genomics).

For analyses of immunoreactivity of variants of the antibody, Ab21 and its mutants were diluted to 1 μM and incubated with 0.16 μM metHb, for 30 min on ice. The binding to autoantigens was assessed by ELISA as described above.

**Size-exclusion chromatography.** Molecular composition of native and metHb-incubated pooled human IgG was assessed by using FPLC Akta Purifier (Cytiva), equipped with Superose 12 10/300 column. Pooled IgG was diluted to 10 μM in PBS and exposed to 2, 4, and 10 μM final concentrations of metHb. In certain cases the treatment was performed in the presence of an excess (5 mM) of KCN. One ml of each sample was loaded on column equilibrated with PBS. The buffer flow rate was set to 0.5 ml/min. Chromatograms were recorded by using UV detection of protein at wavelength of 280 nm.

**Fluorescence spectroscopy.** Fluorescence of hydrophobic probe 8-anilino-1-naphthalenesulfonic acid (ANS) was measured with Hitachi F-2500 fluorescence spectrophotometer (Hitachi Instruments Inc., Wokingham, UK). Pooled human IgG was diluted to 10 μM in PBS and titrated with increasing concentrations of metHb (0, 0.1–10 μM). The samples were incubated for 3 h at 4 °C and then diluted 10× in PBS. Following the dilution, to each sample ANS probe at 10 μM final concentration was added. The emission spectra of ANS following its excitation at 388 nm were recorded in 1 cm quartz cells. The excitation and emission slits were 10 nm. The spectra were recorded in the wavelength range 410–600 nm with a scan speed of 1500 nm/min. All measurements were performed at room temperature.

**Conjugation of hemoproteins to sepharose.** CNBr–activated Sepharose 4B (Cytiva) dry substance was pre-conditioned by sequential washes with deionized $H_2O$ and 1 mM solution of HCl. Equine myoglobulin and cytochrome c were diluted in coupling buffer—0.1 M $NaHCO_3$, 0.5 M NaCl and mixed with the pre-conditioned Sepharose 4B. For each gram dry CNBr–activated Sepharose 4B, a total amount of 40 mg of hemoprotein was used. Following incubation for 1 h, the gel was extensively washed with coupling buffer. Excess of activated groups on the Sepharose was blocked by incubation of the gel with 0.1 M Tris.HCl pH 8 for 1 h at room temperature. To remove non-covalently bound hemoproteins, the gels were washed alternatively (>10 changes) by 0.1 M acetate buffer pH 4, 0.5 M NaCl and 0.1 M Tris.HCl pH 8, 0.5 M NaCl. After this washing, the gels were equilibrated with PBS. Pooled human IgG samples were diluted to 20 μM in PBS and incubated for 3 h at room temperature with sepharose-immobilized hemoproteins (5 ml of IgG solution/2.5 ml of sepharose gel slurry). After the incubation the flow through which contained IgG was collected.

The presence of heme in the flow through fractions was analyzed by a pseudo-peroxidase assay. Briefly, ABTS (Sigma-Aldrich) was diluted at 0.5 mg/ml in citrate-phosphate buffer with pH 5. This solution (890 μl) was mixed with a sample from flow through (100 μl) followed by an addition of $H_2O_2$ to a final concentration of 6 mM. Kinetics of oxidation of ABTS was followed by recording the optical density of solution at 414 nm at intervals of 3 s for a period of 10 min.

ELISA assay was used for evaluation of the immunoreactivities of antibodies that were incubated with hemoprotein-conjugated sepharose.

**Immunoprecipitation.** Sera from healthy blood donors were first diluted 2× in PBS and human metHb added to final concentrations of 0, 5, 10, 20, and 40 μM. In certain cases, the Hb at 100 μM was first pre-treated with 5 mM of KCN. The samples were incubated for 2.5 h at 37 °C with constant shaking.

Magnetic beads with immobilized protein G (Bio-adembeads, Ademtech, Pessac, France) were pre-conditioned by washing with PBS containing 0.65% of Tween 20. Native or metHb exposed sera samples were diluted 5× in PBS containing 0.65% Tween 20 and incubated with the magnetic beads for 75 min at

room temperature. After extensive washing with PBS 0.65% Tween 20 and using of magnetic devise (Adem-Mag) beads were eluted by PAG elution buffer (Ademtech). Eluates were diluted in SDS-PAGE sample buffer (reducing) and subjected to SDS-PAGE using NuPAGE Bis-Tris 10% gels (Novex, Life Technologies, ThermoFisher Scientific). The proteins were transferred to regular nitrocellulose membranes (iBlot gel transfer stacks, Invitrogen, ThermoFisher Scientific) using iBlot electrotransfer system (Invitrogen, ThermoFisher Scientific). After blocking overnight in PBS containing 0.1% Tween 20 and 2% skim milk, the membranes were probed with a goat anti-human Hb IgG (LifeSpan Biosciences, Seattle, WA, USA). After thorough washing with T-TBS, membranes were probed with rabbit anti-goat F(ab)$_2$ fragments conjugates with HRP (human species absorbed, SouthernBiotech, Birmingham, AL, USA). The immunoreactivity was revealed by a chemiluminescence kit (ThermoFisher Scientific) and photo developer (AGFA, Mortsel, Belgium).

**Surface plasmon resonance biosensor analyses.** Kinetics of monoclonal Ab binding to hemoproteins was evaluated by a surface plasmon resonance-based optical biosensor system—Biacore 2000 (Biacore, Cytiva, Uppsala, Sweden). Human metHb, and equine myoglobin (both from Sigma-Aldrich) were covalently immobilized on surface of CM5 sensor chips using an amine-coupling kit (Biacore). Briefly, hemoproteins were diluted in 5 mM maleic acid (pH 3.85) to final concentrations of 25 μg/ml (myoglobin) or 50 μg/ml (metHb) and injected for 4 min over sensor surfaces activated by a mixture of 1-ethyl-3-(3-dimethylaminopropyl)-carbodiimide/N-hydroxysuccinimide (Biacore). The activated carboxyl groups non engaged in interactions with proteins were blocked by exposure to 1 M solution of enthanolamine.HCl, pH 8.5 (Biacore). A control surface was prepared on each chip by activation of the carboxymethylated dextran and subsequent deactivation by 1 M of ethanolamine.HCl. All measurements were performed in HBS-EP (10 mM HEPES pH 7.2; 150 mM NaCl; 3 mM EDTA, and 0.005% Tween 20). The buffer was filtered through 0.22 μm membrane filter and degassed under vacuum.

To evaluate the binding kinetics of the interactions of the monoclonal Ab21 with hemoprotein, the antibody (pre-exposed at 4 μM to 10 μM heme) was serially diluted (twofold dilution step) in HBS-EP to concentrations ranging from 500 to 7.8 nM and injected over the sensor surface. The flow rate during all interaction analyses was set at 25 μl/min. The association and dissociation of the interactions of the heme-exposed monoclonal IgG were monitored for 5 min. The sensor chip surfaces were regenerated by exposure to a solution of 0.1 M glycine pH 12, containing 0.3% Triton-X-100 for a contact time of 60 s. The evaluation of the kinetic data was performed by global kinetic analysis using BIAevaluation version 4.1.1 Software (Biacore). All binding measurements were performed consequently as a function of temperature in the 10–35 °C range. For evaluation of activation and equilibrium thermodynamics, the kinetic data obtained at different temperatures were subjected to standard Eyring analyses as described in ref. [94].

**Statistics and reproducibility.** Most of the presented data were obtained from experiments that were repeated minimum two times. Usually, each particular experiment includes three–four technical replicates. The Kruskal–Wallis statistical test was applied for analyses of some data. Statistical significance was considered as $p \leq 0.05$.

**Reporting summary.** Further information on research design is available in the Nature Portfolio Reporting Summary linked to this article.

## Data availability

The raw data that support the findings of this study are available as Supplementary Data 2, Supplementary Fig. 7, or upon request from the corresponding author.

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

## Acknowledgements

This work was supported by Institut National de la Santé et de la Recherche Médicale (INSERM, France) and by grants from Agence Nationale de la Recherche (ANR-13-JCV1-006-01) and from the European Research Council (Project CoBABATI ERC-StG-678905), both to J.D.D.

## Author contributions

J.D.D. conceived the study; J.D.D., C.P., R.N., L.T.R., S.L.-D., and S.V.K. designed the experiments; C.P., R.N., M.G., M.L., L.T.R., and J.D.D. performed the experiments. C.P., R.N., M.G., M.L., L.T.R., and J.D.D. analyzed the obtained data; C.P., S.L.-D., L.T.R., and J.D.D. contributed to writing and editing of the manuscript.

## Competing interests

The authors declare no competing interests.
