## [Peer Review File · Communications Biology]

Reviewers' comments:

Reviewer #1 (Remarks to the Author):

Hemoglobin (Hb) is a protein in red blood cells that carries oxygen. It comprises four subunits, each having one polypeptide chain (globin part) and one heme group. Heme is the prosthetic group consisting of a protoporphyrin ring and a central iron (Fe) atom. In this work, Planchais et al. document that incubating Hb with human IgGs results in IgG reactivity toward several plasma proteins, including Hb. They propose that such gain of function is due to the transfer of oxidized (Fe³⁺) but not reduced (Fe²⁺) heme from Hb to susceptible IgGs. Finally, they perform binding studies to show that the affinity between a heme-exposed monoclonal IgG towards Hb (and myoglobin) is in the nM range. They conclude that heme-modified IgGs might have a physiological role; they could help haptoglobin eliminate excess Hb from the circulation, thus lowering the toxicity of freely circulating Hb occurring in the context of hemolytic diseases.

Previous studies have established that heme can bind to variable regions of human IgGs, conferring autoreactivity. The same studies have also shown that heme-exposed IgGs recognize a large panel of self- or pathogen-derived antigens with substantial binding affinities (K_d value in the low nanomolar range). Since heme is part of Hb, the authors set up to investigate the impact of Hb on human immunoglobulins. This is because Hb contains heme. Not so surprisingly, the authors found that incubation of polyclonal and some monoclonal human IgGs in the presence of Hb resulted in antibody reactivity towards self-proteins. More interestingly, they found that heme can be transferred from oxidized (Fe³⁺) but not reduced Hb (Fe²⁺) to susceptible IgGs. Unfortunately, however, no data are provided to convincingly prove the transfer mechanism, which would be an exciting new result. Instead, the authors speculate. For example, it is said that reduction of heme in Hb increases its affinity for the globin part, thus diminishing its dissociation rate from Hb. It is also said that the presence of the reducing agents would prevent the ability of heme to oxidize electron-rich amino acid residues in the IgG. Thus, while interesting, I feel the story remains incomplete, especially from the mechanistic standpoint. I also found an unusually high number of typos and some inconsistencies between information reported in materials and methods and figure legends. While these errors can be easily corrected, they lowered my enthusiasm for the work.

Some additional points to consider are:

1) Throughout the manuscript, the authors use Hb and methHb as if these two species were interchangeable. But they are not. Even more so because they try to demonstrate that the oxidation state of heme determines the ability of Hb to induce reactivity against protein targets. I feel that this choice creates much confusion and should be addressed. Based on these considerations, the manuscript title "Hemoglobin triggers antigen-binding polyreactivity of human IgG via transfer of heme" seems misleading, too. Perhaps a better title could be "Oxidized Hemoglobin triggers antigen-binding polyreactivity of human IgG via transfer of heme"

2) Page 5. Line 105. "Previous studies have demonstrated that the exposure of human pooled IgG to heme results in a gain of antibody binding to multiple unrelated protein antigens and to phospholipids". Please provide a reference.

3) Page 5. Line 111. To test Hb-induced IgG autoreactivity, the authors initially used FVIII, FIX, and apo-Hb (globin) as antigens. Subsequently, Ab21 and Rituximab were tested against additional antigens. The rationale for the selection of the antigens, as well as for the selection of Ab21 and Rituximab as model IgGs is unclear and should be better specified.

4) Page 5. Line 116. Where are the supplementary data? I could not find them.

5) Page 5. Line 116. To assess whether Hb (should be methHb) affects IgG binding characteristics in a

"complex" biological milieu, the authors decided to incubate healthy human serum with methHb. They claim that this experimental setting reflects intravascular hemolysis. Why? Serum is the liquid that remains after the clotting of blood. Why should the use of serum reflect intravascular hemolysis? Why not plasma?

6) In Figures 1B, C and G, several bands appear after treatment of polyclonal and monoclonal IgGs with methHb. Despite these are different antibodies, bands are similar, some more intense than others. What are those bands? Also, MWs are missing.

7) Figure 1G shows the difference between control IgG, Ab21, and Rituximab. What is the control IgG sample?

8) In the introduction, Page 4, Line 91, the authors state that a "high level of Hb (20-50 μM) can be maintained in the blood plasma for an extended period of time (>24 hours)". So why is the concentration range of Hb chosen for the experiment in figure 1A only 0-5.12 μM or 0-2.56 μM , depending on whether one reads the figure legend (0-5.12 μM) or material and method section (0-2.56 μM)? How did they come up with the concentrations of IgG and incubation time to be used? Finally, even though figure 1A shows a dose-dependent effect, at higher concentrations of Hb, the response starts to drop. Why?

9) Page 6. Line 122. "The tendency of Hb to induce IgG reactivity to itself was confirmed by fluorescence spectroscopy analyses." This sentence is hard to read. Consider rephrasing.

10) Figure 1E. The fact that the fluorescence signal of ANS for the complex IgG-Hb is lower than the sum of the fluorescence signals measured for the individual proteins is not evidence of binding. Other methods should be used to reach such a conclusion. Also, fluorescent spectra in Figure 1E should be presented after subtraction with the corresponding baseline.

9) Figure 2. Why did the author test IgGs treated with myoglobin and cytochrome C against FIX and apo-Hb and not FVIII or other proteins like in Figure 1? Similarly, why did they test tubulin in Figure 3C?

11) Figure 3. The assumption in this set of experiments is that pre-treatment of Hb with different oxidizing/reducing agents modifies the redox state of heme, hence the ability of Hb to induce autoreactivity. While possible, these reagents could also modify the protein scaffold by introducing PTM, a possibility that should be explored by mass spec. Without mass spec data, I suggest revisiting the text to include limitations and alternative interpretations for these results. Furthermore, the authors said that reducing heme in Hb increases its affinity for the globin part, thus diminishing its dissociation rate from Hb. They also said that the presence of the reducing agents would prevent the ability of heme to oxidize electron-rich amino acid residues in the IgG. Data are needed to substantiate these speculations.

12) Figure 5. Given the complex nature of this analysis, the authors should provide, perhaps in the supplementary materials, a table summarizing association and dissociation rate constants for each experiment, equations used to fit the data, plots of the residuals, and goodness of the fit.

13) The materials and methods section is very hard to read. Consider writing this section more clearly.

Reviewer #2 (Remarks to the Author):

Major point:

This manuscript is interesting enough to provide new insight into the role of heme in antibody polyspecificity. This paper is well organized from heme-mediated induction of novel antigen binding

specificity, to heme transfer, and to the mechanism of antigen recognition. However, the final result section, "Mechanism of heme-induced recognition of hemoproteins by antibodies" does not address the exact mechanism but the property of antigen recognition such as temperature sensitivity or enthalpy and entropy changes. It may be better to ask whether heme itself may make the paratope of the antibody bind to multiple proteins. If authors cannot address this issue by experiments, some detailed discussion is required for the following issues

1. the mechanism of polyspecific alteration of heme-associated antibodies
2. why the oxidative state of heme is essential for polyreactivity
3. whether heme-conjugated non-immunoglobulin proteins can bind multiple proteins

Minor point:

There are a lot of errors in descriptions. Authors need to check erroneous descriptions throughout the manuscript.

- Line 153; I think that this not "cryptic specificities", but "neo-specificities".

- Line 236 & 239; sensitivity → sensitive

Reviewer #3 (Remarks to the Author):

Review report

The manuscript addresses a topic that originated several years ago about posttranslational (redox) modifications of IgG, alterations of their reactivity, specificity, and affinity, and the consequences of these modifications (McIntyre's group: *Throm Res* 2004, *J Autoim* 2005, *Aut Rev* 2005, *Clin Rev Allerg Immunol* 2009...; Rozman/Božič's group: *Aut Rev* 2006, *Ann NY Acad Sci* 2007, *Aut Rev* 2008, *Immunol Immunochem* 2010, *Red REp* 2011, *Lupus* 2015; Dimitrov's group: *J Biol Chem* 2006&2007, *Scand J Immunol* 2007, *Aut Rev* 2008, *Biochem Biophys Res Commun* 2010, *Clin Immunol* 2011, *Med Sci* 2013, *Biochem* 2015, *Inflamm* 2017,...).

Nevertheless, several questions are still unanswered, and some biological roles and mechanisms of the above observations are not well understood. Some of them, especially the role of heme in the situation of pathophysiological increase of plasma hemoglobin in hemolysis, are addressed by the authors of the present manuscript:

- Ill-defined molecular mechanisms responsible for heme-induced diversification of antibody specificity.

- The influence of the early components of hemolysis (Hb) on antibody function.

Research in this area, and thus the manuscript, are important for a better understanding of the functions of extracellular Hb in the context of hemolytic diseases. The authors presented some important information:

About mechanisms:

- transfer of heme from Hb to Ig is indispensable for modulation of Ig reactivity.

- Two previously proposed mechanisms (prooxidant capabilities and interfacial protein bridge) may act synergistically

On the clinical relevance of heme-induced Ab reactivity:

- Prooxidant hemoproteins released in large quantities during hemolysis can induce antibody polyreactivity. This phenomenon may have beneficial effects by facilitating the removal of toxic proteins from the bloodstream.

- The binding properties of a therapeutic antibody, rituximab, may also be altered by the same mechanisms, with potential negative consequences for therapeutic efficacy.

I can fully agree with the authors' last statement, "these findings can serve as a basis for further investigations for understanding of origin of autoantibodies in hemolytic diseases and for better understanding of physiopathology of hemolytic diseases in general. "

There are some minor comments:

1. the study is well supported by the results of various methods used for the analyses to elucidate/illuminate the problem of interactions of Hb with proteins, pooled Ig, MoAb, or therapeutic rituximab. The reader may miss the explanation of why different conditions were used in the phases of this interaction:

Line 365: incubation of heme and Ig with hemoproteins was performed for 30 min on ice (incubation of immunoglobulins with hemoproteins) and Line 409 ...on ice (immunoblot).

Line 385: Pooled IgG and monoclonal antibodies were ... incubated with immobilised proteins for 2 hours at room temperature (ELISA).

Line 460-462: Sera were incubated (with Hb, my comment) for 2.5 hours at 37 °C.

For both protein-protein interactions and redox reactions, temperature is an important factor affecting the rate and range of reactivities/reactions. It is indeed more important for quantitative than for qualitative measurements, but it would be worthwhile to address the reasons and possible consequences of these differences in the discussion.

2. The individual proteins tested (IgG, serum, MoAb, rituximab) were exposed to a considerable range of Hb concentrations, which is important from a methodological point of view and certainly adds value to the experiments. However, the ratios between the Hb concentrations and the concentrations of the tested proteins are quite different. In the methods and materials, the "red line" for such differences (equimolar, excess Hb) is not understood

Lines ...Discussion lines...

3. Line 397: "MoAb at 5 μ M were exposed to 1 μ M for 1h at ice." In the context of the ELISA described, it is not understood to what was MoAb exposed?

4. Line 148 and after: Heme transfer. Very nice experiment on heme transfer (dissociation of globin and association with Ig) by functional assay. It would be interesting to have data from gel filtration/size exclusion chromatography above the expected MM of the heme+Ig complex.

5. Line 153: "We thus investigated whether hemoproteins uncover cryptic specificities". It would be good to explain what kind of "covering" the authors have in mind. There are at least two possibilities: a/ Being covered (and thus uncovered by heme) by part of a "third" non-Ig molecule? In this situation, "uncovering" would mean dissociation of the non-Ig molecule due to the higher affinity of heme to Ig.

b/ To be covered by part of the Ig? In this case, "uncovering" would mean a conformational change of the variable (or even hypervariable) region of Ig.

It would be good to clarify this point.

For conclusion: The statement in line 262 "this is the first demonstration that protein can modify the Ag binding specificity of antibodies posttranslationally" may be true, but the result is expected and understandable. Especially when proteins with coordinatively bound metal ions are used. Fe is bound coordinatively to heme and in direct contact with another protein it acts as a redox-active species. Of course, this point in no way diminishes the significance of the authors' observation. Modifications of immunoglobulin specificities after direct contact of Ig with heme dissociated from Hb at concentrations that can be reached under pathological conditions (hemolysis) is a very important observation. And the presented study is supported by a large number of experiments.

Reviewers' comments:

Reviewer #1 (Remarks to the Author):

Hemoglobin (Hb) is a protein in red blood cells that carries oxygen. It comprises four subunits, each having one polypeptide chain (globin part) and one heme group. Heme is the prosthetic group consisting of a protoporphyrin ring and a central iron (Fe) atom. In this work, Planchais et al. document that incubating Hb with human IgGs results in IgG reactivity toward several plasma proteins, including Hb. They propose that such gain of function is due to the transfer of oxidized (Fe³⁺) but not reduced (Fe²⁺) heme from Hb to susceptible IgGs. Finally, they perform binding studies to show that the affinity between a heme-exposed monoclonal IgG towards Hb (and myoglobin) is in the nM range. They conclude that heme-modified IgGs might have a physiological role; they could help haptoglobin eliminate excess Hb from the circulation, thus lowering the toxicity of freely circulating Hb occurring in the context of hemolytic diseases.

Previous studies have established that heme can bind to variable regions of human IgGs, conferring autoreactivity. The same studies have also shown that heme-exposed IgGs recognize a large panel of self- or pathogen-derived antigens with substantial binding affinities (K_d value in the low nanomolar range). Since heme is part of Hb, the authors set up to investigate the impact of Hb on human immunoglobulins. This is because Hb contains heme. Not so surprisingly, the authors found that incubation of polyclonal and some monoclonal human IgGs in the presence of Hb resulted in antibody reactivity towards self-proteins. More interestingly, they found that heme can be transferred from oxidized (Fe³⁺) but not reduced Hb (Fe²⁺) to susceptible IgGs. Unfortunately, however, no data are provided to convincingly prove the transfer mechanism, which would be an exciting new result. Instead, the authors speculate. For example, it is said that reduction of heme in Hb increases its affinity for the globin part, thus diminishing its dissociation rate from Hb. It is also said that the presence of the reducing agents would prevent the ability of heme to oxidize electron-rich amino acid residues in the IgG. Thus, while interesting, I feel the story remains incomplete, especially from the mechanistic standpoint. I also found an unusually high number of typos and some inconsistencies between information reported in materials and methods and figure legends. While these errors can be easily corrected, they lowered my enthusiasm for the work.

We are grateful to the Reviewer for the comprehensive analyses of our manuscript. We fully agree with the critics that the manuscript presented not sufficient experimental evidence to describe the mechanism of the effect of hemoglobin on human antibodies. To address this issue, we concentrated our efforts to provide further mechanistic details.

Thus, we performed several types of experiments to elucidate details about the transfer of heme from hemoglobin to human IgG. First, we extend our study by using a well characterized human heme-binding monoclonal IgG1, referred to as Ab21. All experiments on hemoglobin effects on IgG polyreactivity were reproduced with this antibody, confirming its suitability for mechanistic investigations. Next, we demonstrated that effect of hemoglobin on Ab21 is considerably inhibited in the presence of hemopexin, a natural scavenger of heme. In contrast, presence of hemopexin that was pre-loaded with heme did not inhibit the effect of hemoglobin. This data clearly demonstrated that the protein free heme is indispensable

for the effect of hemoglobin on human IgG. Further, to directly demonstrate the heme dissociation from hemoglobin and modulation of this dissociation by different conditions, we performed a series of experiments on heme transfer, using absorbance spectroscopy. The obtained data present strong arguments that the oxidized form of heme has decreased affinity for globin and this correlates with capacity for induction of polyreactivity of Ab21 (or pooled human IgG). Finally, to obtain further mechanistic details and directly prove that hemoglobin-derived heme binds to IgG, we performed also site-directed mutagenesis of Ab21. These analyses were guided by molecular modeling of heme interaction with the variable region of the antibody. The site-directed mutagenesis showed that the heme binds to the antigen-binding site of the antibody, using specific tyrosine residues as critical contacts. We also demonstrated that capacity for heme binding and induction of polyreactivity are acquired following somatic hypermutation process of the antibody. Taken together the newly obtained data from these complementary approaches strongly suggest that oxidized hemoglobin liberates free heme which is then bound by the antigen binding site of some antibodies and thus mediates polyreactive antigen binding.

We have now considerably revised the manuscript to incorporate the newly obtained results and to remove some ambiguities. Figure 1 and Figure 2 were merged and rearranged as a new Figure 1. Two additional figures (new Figure 3 and new Figure 5) were added to present newly obtained data. Moreover, we added paragraphs of texts in “Results” and “Discussion” sections of the manuscript, describing the newly obtained results and mechanistic insights. A number of new references have been also added to support our conclusions. All ambiguous and confusing sentences about the mechanism of hemoglobin-induced polyreactivity that were present in the previous version of the manuscript were deleted.

The entire text of the manuscript was edited to correct the typos and other errors as well as to remove the identified discrepancies in the description of technical details of experiments.

Some additional points to consider are:

1) Throughout the manuscript, the authors use Hb and metHb as if these two species were interchangeable. But they are not. Even more so because they try to demonstrate that the oxidation state of heme determines the ability of Hb to induce reactivity against protein targets. I feel that this choice creates much confusion and should be addressed. Based on these considerations, the manuscript title “Hemoglobin triggers antigen-binding polyreactivity of human IgG via transfer of heme” seems misleading, too. Perhaps a better title could be “Oxidized Hemoglobin triggers antigen-binding polyreactivity of human IgG via transfer of heme”

We are grateful to the Reviewer for indicating this imprecision in the manuscript. Indeed, the effect of hemoglobin on human antibodies is exclusively present only if oxidized form of hemoglobin is used. To address this issue, we have now used the abbreviation “metHb” to clearly indicate the use of oxidation form of hemoglobin (or methemoglobin) in the experiments. Moreover, we have changed the title of the revised manuscript to “*Oxidized hemoglobin triggers polyreactivity and autoreactivity of human IgG via transfer of heme*”. This title clearly indicates the role of oxidation status of hemoglobin.

2) Page 5. Line 105. "Previous studies have demonstrated that the exposure of human pooled IgG to heme results in a gain of antibody binding to multiple unrelated protein antigens and to phospholipids". Please provide a reference.

The appropriate references were added.

3) Page 5. Line 111. To test Hb-induced IgG autoreactivity, the authors initially used FVIII, FIX, and apo-Hb (globin) as antigens. Subsequently, Ab21 and Rituximab were tested against additional antigens. The rationale for the selection of the antigens, as well as for the selection of Ab21 and Rituximab as model IgGs is unclear and should be better specified.

To acknowledge this Reviewer's remark, we have now specified the rationale for selection of the antigens and use of the particular antibodies (such as Ab21) in the study.

4) Page 5. Line 116. Where are the supplementary data? I could not find them.

We apologize for not including the Supplemental data file with the previous submission. This file is now enclosed to the submission.

5) Page 5. Line 116. To assess whether Hb (should be metHb) affects IgG binding characteristics in a "complex" biological milieu, the authors decided to incubate healthy human serum with metHb. They claim that this experimental setting reflects intravascular hemolysis. Why? Serum is the liquid that remains after the clotting of blood. Why should the use of serum reflect intravascular hemolysis? Why not plasma?

We agree with Reviewer that plasma in certain cases is more accurate representation of human blood than serum. However, human plasma preparation requires addition of substances that can interfere with heme-induced polyreactivity (EDTA, citric acid) or quench polyreactive antibodies (heparin). This was the reason to select serum for our experiments, where there is no addition of any substance to human blood. Of note, serum is reliable and widely accepted model for complement research, as addition of anti-coagulants interferes with complement system as well. Nevertheless, to acknowledge Reviewer's criticism we edited the claim that addition of hemoglobin to human serum models intravascular hemolysis.

6) In Figures 1B, C and G, several bands appear after treatment of polyclonal and monoclonal IgGs with metHb. Despite these are different antibodies, bands are similar, some more intense than others. What are those bands? Also, MWs are missing.

There is experimental evidence presented in previous studies (PMID #: 33758329; 25742488; 24802758; 17636257) and in the present work (new Figure 5) that some antibodies can bind heme and use its intrinsic protein-binding promiscuity for recognition of unrelated proteins with affinity in nM range. This fact could explain that different antibody molecules demonstrated a similar pattern of recognized proteins when tested for binding to human endothelial cell lysate. In other words, the observed immunoreactivity reflects the intrinsic polyreactivity of heme molecule and specific antibodies serve as a scaffold that display heme in an appropriate manner. We have added a paragraph in the discussion, where we clarified

these issues and discussed putative mechanism of how heme endows antibodies with polyreactivity.

The lack of MW markers in the results presented on Figure 1 is due to the type of gels that were used. In the current study we used single-well ready-made (NuPAGE) gels. These gels are made for loading of a single sample and probing with different samples at later stages of the assay. Since the goal of our experiments was not to identify particular target proteins in the cellular lysates, but to demonstrate overall changes in the immunoreactivity of antibodies upon exposure to oxidized hemoglobin, we consider that the addition of MW is not necessary for demonstrating polyreactivity of antibodies.

7) Figure 1G shows the difference between control IgG, Ab21, and Rituximab. What is the control IgG sample?

The control used in the indicated figure is a human IgG, that is not sensitive to heme.

8) In the introduction, Page 4, Line 91, the authors state that a “high level of Hb (20-50 μ M) can be maintained in the blood plasma for an extended period of time (>24 hours)”. So why is the concentration range of Hb chosen for the experiment in figure 1A only 0-5.12 μ M or 0-2.56 μ M, depending on whether one reads the figure legend (0-5.12 μ M) or material and method section (0-2.56 μ M)? How did they come up with the concentrations of IgG and incubation time to be used? Finally, even though figure 1A shows a dose-dependent effect, at higher concentrations of Hb, the response starts to drop. Why?

Indeed, the concentrations of extracellular hemoglobin reported in hemolytic conditions are much higher as compared in the present study. However, in the complex milieu of human blood there are many additional proteins, lipids, carbohydrates and low molecule compounds which may interfere with the distribution of heme after dissociation from hemoglobin and consequently with its effect on IgG. Moreover, human blood contains a considerable higher concentration of IgG (ca. 10 mg/ml) as well as presence of high concentrations of other immunoglobulin isotypes (IgM and IgA) as compared with selected for in vitro experiments (IgG concentration of 1.5 mg/ml for experiments with pool IgG and 0.3 mg/ml with monoclonal IgG1). The conditions for incubation of human pooled IgG or monoclonal IgG with hemoglobin were selected empirically, following optimization experiments. The incubation time of human IgG with hemoglobin, well reflects the major amounts of heme dissociated from hemoglobin as detected by spectroscopy experiments with hemopexin (presented as new Figure 3B).

Indeed, in many cases incubation with highest concentration of hemoglobin of pooled IgG and monoclonal IgG results in a drop in the dose dependence curve. This drop can be explained by the fact that free hemoglobin (present in solution) when it is at high concentration serves as a target antigen of antibodies, thus blocking their antigen-binding sites and preventing the interaction with surface-immobilized proteins in ELISA experiments. In the revised version of the manuscript, we have now provided this explanation in the “Results” section.

9) Page 6. Line 122. “The tendency of Hb to induce IgG reactivity to itself was confirmed by fluorescence spectroscopy analyses.” This sentence is hard to read. Consider rephrasing.

The indicated sentence was edited to improve its readability.

10) Figure 1E. The fact that the fluorescence signal of ANS for the complex IgG-Hb is lower than the sum of the fluorescence signals measured for the individual proteins is not evidence of binding. Other methods should be used to reach such a conclusion. Also, fluorescent spectra in Figure 1E should be presented after subtraction with the corresponding baseline.

The manuscript used two methodological approaches to provide direct examples for binding of human antibodies to human hemoglobin – ELISA and immunoprecipitation. We consider that the data from ANS fluorescence analyses provided only auxiliary indirect evidence about the interaction. As these data are of less importance, and somehow caused confusion, we have now removed them from Figure 1 and presented them as a Supplemental information. The description of the results was also carefully edited to present these data in a more understandable manner.

9) Figure 2. Why did the author test IgGs treated with myoglobin and cytochrome C against FIX and apo-Hb and not FVIII or other proteins like in Figure 1? Similarly, why did they test tubulin in Figure 3C?

Indeed, in Figure 1, we presented results using a large panel of antigens (particular antigens or human cell lysates). Our rationale was that this figure should depict the induction of polyreactivity and autoreactivity of human pooled and monoclonal IgG, and hence it should be well supported by a large number of examples. In further figures (including in demonstration of the effect of other hemoproteins) we focused mainly to elucidate the mechanism of the phenomenon. Therefore, we considered that the use of two unrelated autoantigens as targets for induction of polyreactivity by hemoglobin is well sufficient for the purpose. We agree with the Reviewer that presenting of tubulin in Figure 3 can confuse the reader. To homogenize the presentation of the data, we have now removed the data about reactivity of antibodies against tubulin from Figure 3.

11) Figure 3. The assumption in this set of experiments is that pre-treatment of Hb with different oxidizing/reducing agents modifies the redox state of heme, hence the ability of Hb to induce autoreactivity. While possible, these reagents could also modify the protein scaffold by introducing PTM, a possibility that should be explored by mass spec. Without mass spec data, I suggest revisiting the text to include limitations and alternative interpretations for these results. Furthermore, the authors said that reducing heme in Hb increases its affinity for the globin part, thus diminishing its dissociation rate from Hb. They also said that the presence of the reducing agents would prevent the ability of heme to oxidize electron-rich amino acid residues in the IgG. Data are needed to substantiate these speculations.

We agree with the Reviewer that some of the agents used in the study (especially hydrogen peroxide and sodium dithionite) can induce post-translational modifications to IgG and/or hemoglobin molecules. This is one of the reasons to apply protein scavenger of heme, hemopexin, in our additional experiments, demonstrating the effect of heme transfer from hemoglobin on the polyreactivity of antibodies.

To address the Reviewer's concern, we have now added a sentence in the revised manuscript, acknowledging the potential presence of post-translational modifications in the studied proteins. As based on our novel data (described above) and the effect of ROS scavenging enzymes and deuterium oxide (see Supplemental Figure 6) we have now ruled out the hypothesis that a major hemoglobin-driven oxidation of IgG molecules plays a central role for the observed induction of polyreactivity and autoreactivity of antibodies. We have now revised the text of the manuscript and removed this speculation.

12) Figure 5. Given the complex nature of this analysis, the authors should provide, perhaps in the supplementary materials, a table summarizing association and dissociation rate constants for each experiment, equations used to fit the data, plots of the residuals, and goodness of the fit.

A supplementary table with additional information about the kinetics analyses has been now submitted along with the manuscript (Supplemental Table 1).

13) The materials and methods section is very hard to read. Consider writing this section more clearly.

To address this Reviewer's criticism, we have edited "*Material and Methods*" section. In addition, we added to this part new pieces of text describing the novel experimental approaches.

Reviewer #2 (Remarks to the Author):

Major point:

This manuscript is interesting enough to provide new insight into the role of heme in antibody polyspecificity. This paper is well organized from heme-mediated induction of novel antigen binding specificity, to heme transfer, and to the mechanism of antigen recognition. However, the final result section, "Mechanism of heme-induced recognition of hemoproteins by antibodies" does not address the exact mechanism but the property of antigen recognition such as temperature sensitivity or enthalpy and entropy changes. It may be better to ask whether heme itself may make the paratope of the antibody bind to multiple proteins. If authors cannot address this issue by experiments, some detailed discussion is required for the following issues

We are grateful to the Reviewer for the positive evaluation of our work and for the constructive remarks about the interpretation of the data. We agree that the study of the kinetics and thermodynamics of interaction of Ab21 with hemoprotein did not provide direct mechanistic details about the hemoglobin effect on antibodies. In the revised version of the manuscript, we have now removed these claims.

1. the mechanism of polyspecific alteration of heme-associated antibodies

We performed a series of experiments aiming to better demonstrate the transfer of heme from hemoglobin to IgG as well as to elucidate how heme modifies the antigen-binding specificity of human antibodies (see answer to Reviewer 1). The newly obtained data from

site-directed mutagenesis analyses helped us to characterize the molecular requirement for heme binding to human IgG and for induction of polyreactivity. We identified that specific residues (Tyr) in the CDR H3 of the variable region of heavy chain have a critical importance for hemoglobin-induced polyreactivity of antibodies. These residues are also known to play an important function for heme binding. The newly obtained data have been now presented in the revised manuscript as two new figures (Figure 3 and 5), and described in "Results" section. Based on these newly added results and the data from previous studies, we have also extended and refined our discussion about the mechanism of heme induced polyreactivity and autoreactivity of human antibodies.

2. why the oxidative state of heme is essential for polyreactivity

There are several reports (as for example PMID: 27642551) in the literature where the dissociation of heme from hemoglobin (or myoglobin) was elucidated as the function of the redox status of heme's iron ion. These reports demonstrated that the reduced form of heme (ferrous, Fe^{2+}) has a considerably higher affinity for the polypeptide chains of the hemoproteins than the oxidized (ferric, Fe^{3+}) form. Based on these data and the results obtained in our study, we concluded that the redox status of heme's iron is of importance for induction of polyreactivity by controlling the rate of transfer of the prosthetic group from hemoglobin towards antibodies. These aspects have been now discussed in the revised version of the manuscript.

3. whether heme-conjugated non-immunoglobulin proteins can bind multiple proteins

We are thankful for this constructive remark. Indeed, heme has been shown to bind to many human proteins. We believe that if displayed in a suitable manner, i.e. exposed to the solvent surface, heme may serve as an interfacial cofactor for binding of protein to other proteins. To the best of our knowledge, apart for immunoglobulins there are not research studies for identification of such interactions mediated by heme.

Minor point:

There are a lot of errors in descriptions. Authors need to check erroneous descriptions throughout the manuscript.

- Line 153; I think that this not "crptic specificities", but "neo-specificities".

- Line 236 & 239; sensitivity → sensitive

We carefully checked the text of the manuscript and corrected identified mistakes.

Reviewer #3 (Remarks to the Author):

Review report

The manuscript addresses a topic that originated several years ago about posttranslational (redox) modifications of IgG, alterations of their reactivity, specificity, and affinity, and the consequences of these modifications (McIntyre's group: Throm Res 2004, J Autoim 2005, Aut Rev 2005, Clin Rev Allerg Immunol 2009...; Rozman/Božič's group: Aut Rev 2006, Ann NY Acad Sci 2007, Aut Rev 2008, Immunol Immunochem 2010, Red REp 2011, Lupus 2015; Dimitrov's

group: J Biol Chem 2006&2007, Scand J Immunol 2007, Aut Rev 2008, Biochem Biophys Res Commun 2010, Clin Immunol 2011, Med Sci 2013, Biochem 2015, Inflamm 2017,...).

Nevertheless, several questions are still unanswered, and some biological roles and mechanisms of the above observations are not well understood. Some of them, especially the role of heme in the situation of pathophysiological increase of plasma hemoglobin in hemolysis, are addressed by the authors of the present manuscript:

- Ill-defined molecular mechanisms responsible for heme-induced diversification of antibody specificity.

- The influence of the early components of hemolysis (Hb) on antibody function.

Research in this area, and thus the manuscript, are important for a better understanding of the functions of extracellular Hb in the context of hemolytic diseases. The authors presented some important information:

About mechanisms:

- transfer of heme from Hb to Ig is indispensable for modulation of Ig reactivity.

- Two previously proposed mechanisms (prooxidant capabilities and interfacial protein bridge) may act synergistically

On the clinical relevance of heme-induced Ab reactivity:

- Prooxidant hemoproteins released in large quantities during hemolysis can induce antibody polyreactivity. This phenomenon may have beneficial effects by facilitating the removal of toxic proteins from the bloodstream.

- The binding properties of a therapeutic antibody, rituximab, may also be altered by the same mechanisms, with potential negative consequences for therapeutic efficacy.

I can fully agree with the authors' last statement, "these findings can serve as a basis for further investigations for understanding of origin of autoantibodies in hemolytic diseases and for better understanding of physiopathology of hemolytic diseases in general. "

We are grateful to the Reviewer for the comprehensive analysis of our study and the positive evaluation.

There are some minor comments:

1. the study is well supported by the results of various methods used for the analyses to elucidate/illuminate the problem of interactions of Hb with proteins, pooled Ig, MoAb, or therapeutic rituximab. The reader may miss the explanation of why different conditions were used in the phases of this interaction:

Line 365: incubation of heme and Ig with hemoproteins was performed for 30 min on ice (incubation of immunoglobulins with hemoproteins) and Line 409 ...on ice (immunoblot).

Line 385: Pooled IgG and monoclonal antibodies were ... incubated with immobilised proteins for 2 hours at room temperature (ELISA).

Line 460-462: Sera were incubated (with Hb, my comment) for 2.5 hours at 37 °C.

For both protein-protein interactions and redox reactions, temperature is an important factor affecting the rate and range of reactivities/reactions. It is indeed more important for quantitative than for qualitative measurements, but it would be worthwhile to address the reasons and possible consequences of these differences in the discussion.

Indeed, the experimental setting in the manuscript have some discrepancies in incubation times and conditions. However, these discrepancies originated from rational reasons. Thus, for all immunoassays, the antibodies were incubated with hemoglobin for 30

min on ice. We found that at this condition there is a sufficiently high effect of hemoglobin on human IgG. In our preliminary optimization experiments, we performed time-dependence of effect of hemoglobin on antibodies by measuring the effect on immunoreactivity as a function of incubation time (varying from 0 to 4 hours). These experiments demonstrated that the incubation of human antibodies for 30 min with hemoglobin results in a considerable augmentation of the reactivity of human antibodies. For practical reasons and due to high sensitivity of immunoassays, we selected this condition for most of the experiments presented in this study.

Longer times of incubation and different temperatures were used only when we tested the effect of heme in a complex biological milieu (human serum) or when hemoproteins were immobilized on matrix. In the first case the reason was that presence of other proteins can compete with the immunoglobulins for interaction with free heme. In the second setting long incubation time was selected to ensure detection of heme bound on IgG molecules (see Figure 2A), for which a pseudoperoxidase catalytic assays was used. This assay requires large concentrations of samples (especially of protein-bound heme) as compared to the immunoassays.

In the revised version of the manuscript, we have now added information about the rationale to use the selected experimental conditions.

2. The individual proteins tested (IgG, serum, MoAb, rituximab) were exposed to a considerable range of Hb concentrations, which is important from a methodological point of view and certainly adds value to the experiments. However, the ratios between the Hb concentrations and the concentrations of the tested proteins are quite different. In the methods and materials, the "red line" for such differences (equimolar, excess Hb) is not understood

We used a broad range of hemoglobin concentrations for treatment of human immunoglobulin preparations to ensure the detection of concentration-dependencies of effects. This warrants the reliability and better comparison of the obtained data, especially when the influence of different substances is compared. The concentration of pooled IgG (IVIG) and monoclonal antibodies, (10 and 2 μ M, respectively) were selected based on technical reasons. The fact that pooled IgG preparation was used at 5-folds higher concentration than the monoclonal IgG1 antibody is because it represents a heterogenous mixture of IgGs and contains only a fraction of antibodies that can acquire polyreactivity after exposure to heme. In earlier studies it was estimated that this fraction ranges between 10 and 20 % of total antibodies. Thus, we decided to use it at 5-fold higher concentration as compared with the selected human monoclonal IgG1 that demonstrated a high sensitivity to heme and served as a model.

Lines ...Discussion linjes...

3. Line 397: "MoAb at 5 μ M were exposed to 1 μ M for 1h at ice." In the context of the ELISA described, it is not understood to what was MoAb exposed?

We have now corrected this error.

4. Line 148 and after: Heme transfer. Very nice experiment on heme transfer (dissociation of globin and association with Ig) by functional assay. It would be interesting

to have data from gel filtration/size exclusion chromatography above the expected MM of the heme+Ig complex.

We agree with the Reviewer that data about changes in the molecular weight of IgG pre-incubated with hemoglobin would be a valuable prove for the presence of heme transfer. Unfortunately, generation of these data represents a technical challenge as the molecular weight of heme is 652 Da and this of IgG is approximately 150 000 Da. With analytical techniques that we have access, such as size exclusion chromatography, the difference could not be detected. Moreover, in the experiments for direct heme transfer from hemoproteins (Figure 2) referred by Reviewer, we used a pooled IgG preparation, this preparation is highly heterogenous and the MW of IgG molecules that constitute it may vary by >1000 Da due to differences in the glycosylation patterns and sequences of variable regions.

5. Line 153: "We thus investigated whether hemoproteins uncover cryptic specificities". It would be good to explain what kind of "covering" the authors have in mind. There are at least two possibilities:

a/ Being covered (and thus uncovered by heme) by part of a "third" non-Ig molecule? In this situation, "uncovering" would mean dissociation of the non-Ig molecule due to the higher affinity of heme to Ig.

b/ To be covered by part of the Ig? In this case, "uncovering" would mean a conformational change of the variable (or even hipervariable) region of Ig.

It would be good to clarify this point.

We are grateful to the Reviewer for these constructive insights. Based on the newly obtained data and previous studies from our laboratory, we consider that the second possible mechanism is the more relevant one. To acknowledge the Reviewer's concern, we have now elaborated on the subject on the mechanism of heme-induced polyreactivity. A paragraph discussing the mechanism was added at an appropriate junction in the "Discussion" of the manuscript.

For conclusion: The statement in line 262 "this is the first demonstration that protein can modify the Ag binding specificity of antibodies posttranslationally" may be true, but the result is expected and understandable. Especially when proteins with coordinatively bound metal ions are used. Fe is bound coordinatively to heme and in direct contact with another protein it acts as a redox-active species.

The text of the manuscript has been now edited to acknowledge this Reviewer's remark.

Of course, this point in no way diminishes the significance of the authors' observation. Modifications of immunoglobulin specificities after direct contact of Ig with heme dissociated from Hb at concentrations that can be reached under pathological conditions (hemolysis) is a very important observation. And the presented study is supported by a large number of experiments.

We are grateful for these very positive concluding remarks by the Reviewer.

Reviewers' comments:

Reviewer #1 (Remarks to the Author):

I appreciate that the authors provided lots of new data to substantiate their hypotheses, strengthening their claims. While I am satisfied with most of their answers, I am still concerned about the presentation of their findings. To me, the result section of the manuscript reads as a collection of experiments, which is extremely difficult to follow. This also shows in the abstract. For example, if one of the significant points is that human antibodies acquire polyreactivity and autoreactivity after incubation with oxidized and not reduced Hb, why not address this question at the very beginning and then describe the mechanism? Interestingly, the abstract does not mention that Hb can adopt oxidized and reduced states, which needs to be clarified for a reader with limited knowledge in this field. While, in principle, I support publication, I strongly suggest reorganizing the result section to provide a more cohesive narrative to improve the findings' readability and impact.

Some other minor comments:

- 1) Figure 1g shows a blot with and without KCN. However, the rationale for this experiment is not discussed in the text. Since cyanide anions represent a high-affinity ligand for heme's iron ion, I suspect KCN was used to demonstrate that this process requires iron ions, which they show later. This goes back to my original comment that it takes much effort to follow how the results are presented.
- 2) Figure 3d. "...Absorbance spectra of xxxx. The spectra were taken at intervals of 30 min for total time of 8 hours". It is unclear which one is time 0 and which one is time 8 hours.
- 3) There still are many sentences in which Hb is used instead of metHb, which needs to be corrected. For example: on page 5, line 99: "... We demonstrate that incubation of polyclonal or some monoclonal human IgG antibodies in the presence of hemoglobin resulted in an appearance of polyreactivity and reactivity towards distinct self-proteins". page 6, line 135, "The effect of Hb on the antigen-binding reactivity of the Ab21 was concentration-dependent as observed by immunoblot 137 analyses (Fig. 1c).

Reviewer #2 (Remarks to the Author):

The addressed issues were satisfactorily discussed with additional experiments.

Reviewer #3 (Remarks to the Author):

The authors responded to my comments on two ways:

- by including additional information, data and explanation in the new version of the manuscript (mostly)

- by explanation in the letter (minimally)

They fulfilled my expectations and I have no additional comment to the new version of the manuscript

Reviewer #1:

I appreciate that the authors provided lots of new data to substantiate their hypotheses, strengthening their claims. While I am satisfied with most of their answers, I am still concerned about the presentation of their findings. To me, the result section of the manuscript reads as a collection of experiments, which is extremely difficult to follow. This also shows in the abstract. For example, if one of the significant points is that human antibodies acquire polyreactivity and autoreactivity after incubation with oxidized and not reduced Hb, why not address this question at the very beginning and then describe the mechanism? Interestingly, the abstract does not mention that Hb can adopt oxidized and reduced states, which needs to be clarified for a reader with limited knowledge in this field. While, in principle, I support publication, I strongly suggest reorganizing the result section to provide a more cohesive narrative to improve the findings' readability and impact.

We are thankful to the Reviewer for appreciating our revision effort. We are also grateful for the critical remarks about the logic of presentation of results in the manuscript. We completely agree that the presentation of the results about the importance of oxidation state of heme in hemoglobin makes more sense to come before presentation of the results about the transfer of heme. To address this issue, we have now reorganized the Results section of the manuscript. The order of figures has been changed – the previous version Figure 4 became in the revised manuscript Figure 2. We also changed accordingly the order of presentation of Supplemental figures.

In addition, the abstract was edited to clearly indicate that the effects on antibodies were observed only with methemoglobin.

Some other minor comments:

1) Figure 1g shows a blot with and without KCN. However, the rationale for this experiment is not discussed in the text. Since cyanide anions represent a high-affinity ligand for heme's iron ion, I suspect KCN was used to demonstrate that this process requires iron ions, which they show later. This goes back to my original comment that it takes much effort to follow how the results are presented.

In the revised version of the manuscript, we have provided an explanation about the rationale for use of CN anions in this specific experiment. A fragment of text describing the effect of cyanide anions on heme was transferred from further parts of the manuscript to the specific juncture.

2) Figure 3d. "...Absorbance spectra of xxxx. The spectra were taken at intervals of 30 min for total time of 8 hours". It is unclear which one is time 0 and which one is time 8 hours.

To improve the readability of the figure we have edited it and presented the first spectroscopic reading as a thick red line. The last reading (at 8 hours) is presented with blue thick line but this line is barely visible as most of the lines obtained at earlier time points also clustered together and masked it. In the revised version of the manuscript, we have provided explanation in the figure legend.

3) There still are many sentences in which Hb is used instead of metHb, which needs to be corrected. For example: on page 5, line 99: "... We demonstrate that incubation of polyclonal or some monoclonal human IgG antibodies in the presence of hemoglobin resulted in an appearance of polyreactivity and reactivity towards distinct self-proteins". page 6, line 135, "The effect of Hb on the antigen-binding reactivity of the Ab21 was concentration-dependent as observed by immunoblot 137 analyses (Fig. 1c).

The indicated mistakes as well as other similar found in the text have now been corrected.

Reviewer #2:

The addressed issues were satisfactorily discussed with additional experiments.

Reviewer #3:

The authors responded to my comments on two ways:

- by including additional information, data and explanation in the new version of the manuscript (mostly)
- by explanation in the letter (minimally)

They fulfilled my expectations and I have no additional comment to the new version of the manuscript

We are grateful to the Reviewers for positive assessment of our revision.